# Multi-layered genetic approaches to identify approved drug targets

## Graphical abstract

## Authors

Marie C. Sadler, Chiara Auwerx,
Patrick Deelen, Zoltán Kutalik

## Correspondence

zoltan.kutalik@unil.ch

## In brief

Sadler et al. compared and benchmarked genetically informed approaches combined with network diffusion to prioritize drug target genes. Gene prioritization methods were based on large-scale genetic studies such as genome-wide association and whole-exome studies, as well as tissue-wide and whole-blood expression and protein quantitative trait loci.

## Highlights

- Gene prioritization methods based on GWAS, exome, and QTL data

- Comparison of methods for drug target identification across 30 clinical traits

- We found a 1.3- to 2.2-fold enrichment for drug targets among prioritized genes

- Network diffusion of prioritized genes significantly boosted performance

 Sadler et al., 2023, Cell Genomics 3, 100341
July 12, 2023 © 2023 The Author(s).

# Cell Genomics

CellPress

## Article

# Multi-layered genetic approaches to identify approved drug targets

Marie C. Sadler,[1,2,3] Chiara Auwerx,[1,2,3,4] Patrick Deelen,[5,6] and Zoltán Kutalik[1,2,3,7,*]

[1]University Center for Primary Care and Public Health, Route de Berne 113, 1010 Lausanne, Switzerland
[2]Swiss Institute of Bioinformatics, Quartier Sorge, 1015 Lausanne, Switzerland
[3]Department of Computational Biology, University of Lausanne, 1015 Lausanne, Switzerland
[4]Center for Integrative Genomics, University of Lausanne, 1015 Lausanne, Switzerland
[5]Department of Genetics, University of Groningen, 9700 Groningen, the Netherlands
[6]Oncode Institute, 3521 Utrecht, the Netherlands
[7]Lead contact
*Correspondence: zoltan.kutalik@unil.ch

## SUMMARY

Drugs targeting genes linked to disease via evidence from human genetics have increased odds of approval. Approaches to prioritize such genes include genome-wide association studies (GWASs), rare variant burden tests in exome sequencing studies (Exome), or integration of a GWAS with expression/protein quantitative trait loci (eQTL/pQTL-GWAS). Here, we compare gene-prioritization approaches on 30 clinically relevant traits and benchmark their ability to recover drug targets. Across traits, prioritized genes were enriched for drug targets with odds ratios (ORs) of 2.17, 2.04, 1.81, and 1.31 for the GWAS, eQTL-GWAS, Exome, and pQTL-GWAS methods, respectively. Adjusting for differences in testable genes and sample sizes, GWAS outperforms e/pQTL-GWAS, but not the Exome approach. Furthermore, performance increased through gene network diffusion, although the node degree, being the best predictor (OR = 8.7), revealed strong bias in literature-curated networks. In conclusion, we systematically assessed strategies to prioritize drug target genes, highlighting the promises and pitfalls of current approaches.

## INTRODUCTION

Drugs whose targets have genetic support were found to be more likely to succeed in clinical trials.[1,2] Although multiple methods have been proposed to establish such genetic support, leveraging genetic data to find disease genes, and ultimately drug targets, has proven to be challenging.[3–6] The most straightforward approach maps genome-wide association study (GWAS) signals to the closest genes, with more sophisticated methods incorporating linkage disequilibrium (LD) structure and gene annotation information to compute gene scores.[7–9] Over the past decade, large-scale molecular quantitative trait loci (mQTL) datasets facilitated the discovery of disease mechanisms and the identification of potential new drug targets.[10–15] Several methods, including Mendelian randomization studies, transcriptome-wide association studies, and colocalization methods have integrated expression and protein QTL data with GWASs to pinpoint likely causal genes for complex traits and diseases.[16–22] More recently, the availability of high-throughput sequencing data enabled the discovery and analysis of rare variants and their aggregated effects to reveal gene-disease associations.[23,24] Whole-exome sequencing (WES) in the UK Biobank (UKBB) showed that genes prioritized this way are 3.6 times more likely to be targets of drugs approved by the US Food and Drug Administration (US FDA).[25]

Genes prioritized by GWASs, mQTL-GWAS integration methods, and WES burden tests may not be drug targets themselves, but may be up- or downstream of those in pharmacological pathways. Propagating gene prioritization scores on networks has proven to be a promising approach to identify known drug target genes.[26–30] Starting from seed genes (i.e., prioritized disease-associated genes), network connectivity can identify neighboring genes that strongly interact with disease genes, but lack direct genetic evidence that explains their therapeutic effect. Gene networks can be derived from literature or high-throughput experiments and thus are prone to yielding very different results when used for (seed) gene score diffusion.[31]

Here, we took a comprehensive approach to examine the contribution of each method component to the success of drug target prioritization. First, we focused on four different approaches to prioritize (seed) genes: (1) LD-aware gene score computation from the largest GWASs with full publicly available summary statistics (Pascal[9]); (2) Mendelian randomization (MR) combining tissue-wide expression QTLs and GWASs (eQTL-GWAS); (3) MR combining plasma protein QTL with GWAS (pQTL-GWAS); and (4) UKBB WES burden tests (Exome). We then used three different networks to diffuse the seed gene scores: (1) the STRING protein-protein interaction (PPI) network[32]; (2) an RNA-sequencing (RNA-seq) coexpression network[33]; and (3) the FAVA network.[34] All 12 combinations of

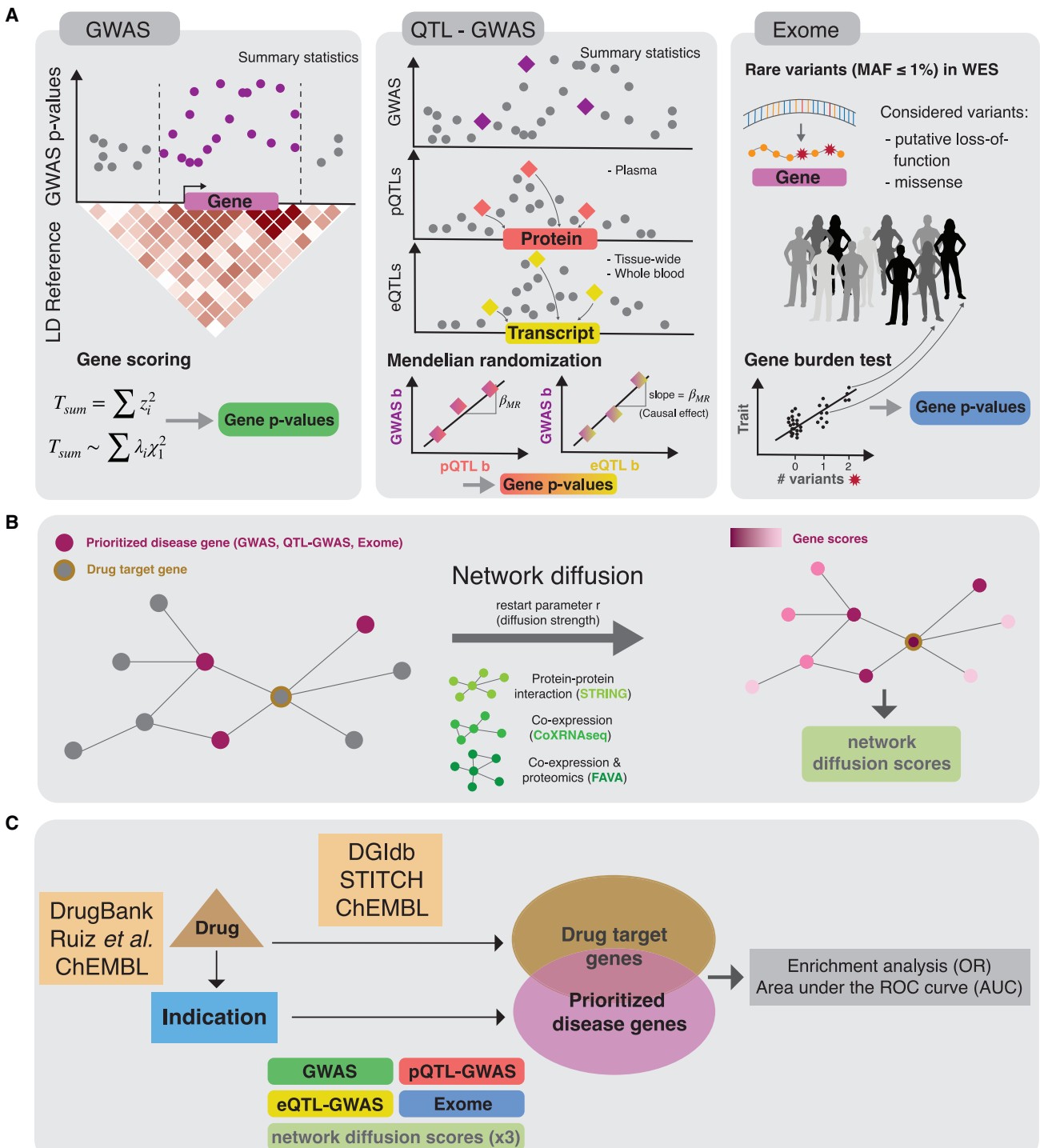

**Figure 1. Overview of the analysis workflow**

(A) Three different gene prioritization methods were tested in this study. The first one uses GWAS summary statistics as input (GWAS). The second combines molecular QTL and GWAS summary statistics (QTL-GWAS): either expression QTL (eQTL) or protein QTL (pQTL) data. The third leverages individual-level whole-exome sequencing (WES) data (Exome). In the GWAS method, gene p values are based on the sum of squared SNP $Z$ scores ($T_{sum}$) that follows a weighted $\chi_1^2$ distribution. The QTL-GWAS method integrates QTL and GWAS summary statistics through Mendelian randomization (MR). MR causal effect sizes ($\beta_{MR}$) are calculated from GWAS and mQTL effect sizes (GWAS b and mQTL b, respectively) and gene scores are the corresponding p values. The Exome method aggregates rare variants from WES data. Putative loss-of-function and missense variants with minor allele frequencies (MAF) below 1% are collapsed in burden tests, which results in gene p values. The different approaches were benchmarked for their ability to prioritize drug target genes.

*(legend continued on next page)*

the four seed-generating methods and the three networks were applied to 30 traits (Figure 1) using five different reference sets of target genes (DrugBank,[35] Ruiz et al.,[36] ChEMBL,[37] DGIdb,[38] and STITCH[39]). Overall, we provide an in-depth comparison of all combinations of these approaches, identifying their respective strengths and caveats.

## RESULTS

### Overview of the analysis

In this study, we calculated gene prioritization scores and tested their ability to identify drug targets across 30 traits (Figure 1). We focused on three types of method, termed GWAS, QTL-GWAS, and Exome, that allow the computation of gene scores provided genetic association data (Figure 1A).

The GWAS method takes as input GWAS summary statistics together with a matching LD reference panel. Gene p values are calculated based on the sum of squared test statistics for SNPs falling into the gene region.[9] The QTL-GWAS methods integrate GWAS summary statistics with mQTL data for the gene of interest. We calculated gene scores using (1) eQTL-GWAS data from the largest available whole-blood eQTL study (eQTLGen study, n = 31,684)[13] as well as tissue-wide eQTL data from the GTEx Consortium v.8 (n = 65–573 for 48 tissue types)[40] and (2) pQTL-GWAS data from the largest available plasma pQTL study (deCODE study, n = 35,559).[14] Integration was done by performing MR analyses using either the protein or the transcript as exposure and the GWAS trait as outcome. If not specified otherwise, the eQTL-GWAS method refers to the tissue-wide analysis in which the eQTLGen and GTEx data are combined by considering the tissue for which the MR effect was the most significant (STAR Methods). While the GWAS and QTL-GWAS methods focus on common genetic variants, the Exome method considers only rare variants from WES data with minor allele frequencies (MAFs) below 1%. Gene scores were based on gene burden tests that aggregate putative loss-of-function and missense variants, and we used the resulting p values from the WES analysis in the UKBB.[25] To allow for a fair comparison with the Exome method while also exploiting disease-specific consortium GWAS summary statistics with maximized case counts, we calculated gene prioritization scores for the GWAS and QTL-GWAS methods using both consortium GWAS and UKBB GWAS data that matched Exome sample sizes (Tables S1 and S2; STAR Methods).

Disease genes may not coincide with drug target genes, but they may be in close proximity in terms of molecular interaction (Figure 1B). Through diffusion based on random walks, we leveraged network connectivity to prioritize neighbors of disease genes, which may be drug targets. We tested this hypothesis on three different network types: the STRING PPI network, which relies on literature interactions, among other data types[32]; a gene

coexpression network based on 31,499 RNA-seq samples (CoXRNAseq)[33]; and a gene coexpression network based on single-cell RNA-seq and proteomics data (FAVA).[34] Gene prioritization scores were obtained following diffusion at six different restart parameter values (r = 0, 0.2, 0.4, 0.6, 0.8, 1) (STAR Methods).

Disease drug target genes were defined using public databases. Specifically, drug-disease indications were retrieved from DrugBank,[35] Ruiz et al.,[36] and ChEMBL,[37] while drug-drug target pairs originated from DGIdb,[38] STITCH,[39] and ChEMBL.[37] Drug target enrichment analyses were calculated for the following five database combinations: DrugBank/DGIdb, DrugBank/STITCH, Ruiz/DGIdb, Ruiz/STITCH, and ChEMBL/ChEMBL.

Finally, prioritized disease genes, defined as the top 1% of genes identified through the 12 combinations of gene prioritization and network diffusion methods (5% for combinations involving the pQTL-GWAS method to account for the smaller set of testable genes), were then tested for enrichment with the five drug target genes using Fisher's exact test (Figure 1C). Background genes were defined as all genes that could be tested by the respective method, and sensitivity analyses were performed on background genes testable for all methods. Second, we calculated the area under the receiver operating characteristic curve (AUC) values, which has the advantage of not requiring any thresholds. To compute a combined enrichment score per method, we aggregated results across traits and drug databases termed overall odds ratios (ORs) or overall AUC values (STAR Methods).

### Concordance of prioritized genes among gene scoring methods

We first analyzed whether genes prioritized by the GWAS, QTL-GWAS, and Exome methods were concordant (Figure 2). For each of the 30 traits, we calculated gene scores for the testable autosomal protein-coding genes (GWAS, ~19,150; eQTL-GWAS, ~12,550 (blood) and ~16,250 (tissue-wide); pQTL-GWAS, ~1,870; Exome, ~18,800). In the tissue-wide eQTL-GWAS method, the tissue with the most significant MR p value was selected. In Figure S1, we show the proportion of genes mapped to a particular tissue category. The contributions of glandular-endocrine, neural central nervous system (CNS), and whole-blood (eQTLGen) tissue categories were the highest (respective means of 15.3%, 12.8%, and 12.6% across the 30 traits; Tables S3 and S4). Although each trait had genes mapped to nearly all tissues, a few distinctive patterns could be observed: cardiac muscle tissues contributed the most to atrial fibrillation (16.4%); vascular tissues the most to coronary artery disease (16.5%), followed by diastolic (11.1%) and systolic (9.9%) blood pressure; and the neural CNS the most to schizophrenia (16.9%) and bipolar disease (16.6%).

(B) The effects of network diffusion using three different network types and different diffusion strengths (i.e., restart parameter *r*) were evaluated. Drug target genes may be prioritized only following signal propagation from neighboring disease genes.

(C) Diseases were linked to target genes through public drug databases: first, we used drug-indication information to connect the 30 traits to drugs and then leveraged drug target information to link the drugs to genes. Prioritized disease genes and corresponding diffusion scores (obtained via strategies described in A and B) were then tested for overlap with drug target genes through Fisher's exact test, resulting in odds ratios (ORs), and through area under the receiver operating characteristic curve (AUC) values.

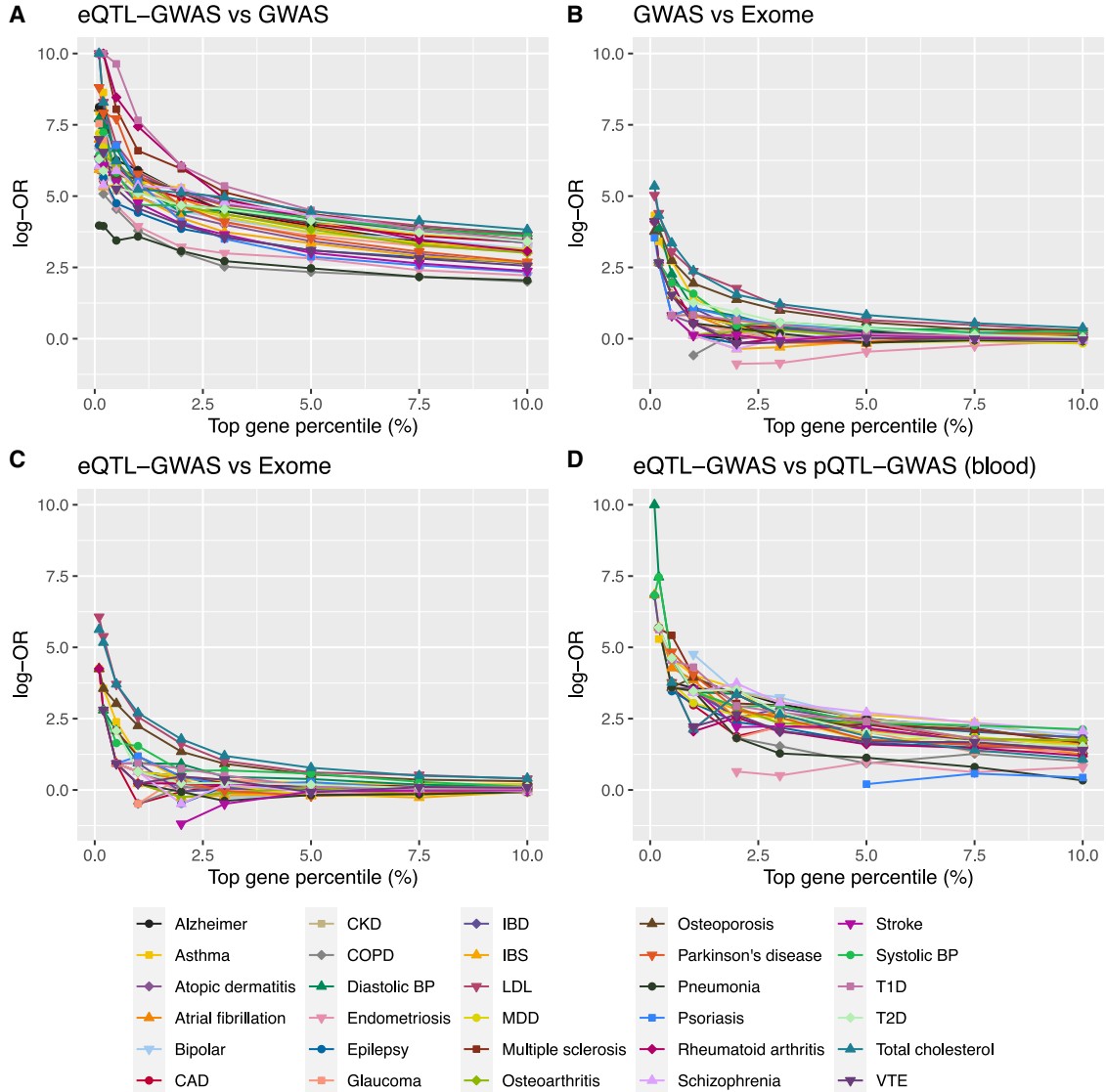

**Figure 2. Evaluating the concordance of prioritized genes among gene scoring methods**

(A–D) The top prioritized genes between pairs of methods were compared at different thresholds for each of the 30 traits/drug indications. The logarithm of odds ratios (log-OR) was calculated from Fisher's exact tests. Log-ORs are plotted only for percentiles at which common genes between pairs of methods were found. Comparisons were conducted on the same background genes and same data origins (i.e., on UK Biobank GWASs for comparisons with the Exome method). Tissue-wide eQTL-GWAS gene prioritizations were considered for the comparison with the GWAS and Exome methods and the blood-only eQTL-GWAS gene prioritization method for the comparison with the pQTL-GWAS method.

The concordance of prioritized genes among pairs of methods is summarized in Figure 2. For each trait, we calculated Fisher's exact tests between the top prioritized genes at thresholds ranging in the top 0.1%–10% (STAR Methods). The overlap was the highest between the GWAS and the eQTL-GWAS methods (Figure 2A). At 1%, the median OR was 212.2, which dropped to 51.0 and 22.1 at 5% and 10%, respectively. The overlap of prioritized genes was the lowest with the Exome method. The top 1% GWAS vs. Exome and eQTL-GWAS vs. Exome overlaps (based only on UKBB GWAS summary statistics), yielded median ORs of 1.7 and 1.9, respectively, which dropped to 1.0 at 10% for both methods (Figures 2B and 2C).

Median ORs between eQTL-GWAS (whole blood) and pQTL-GWAS (blood plasma) were 8.5 and 4.6 at the top 5% and 10%, respectively (Figure 2D).

### Enrichment of prioritized genes for drug targets

Next, we assessed the extent to which prioritized genes overlapped with drug target genes. For each trait, we conducted enrichment analyses for the GWAS, eQTL-GWAS, pQTL-GWAS, and Exome methods using our five definitions of drug target genes.

In Figure 3A, we show the resulting ORs for the DrugBank/DGIdb database combination. Across methods, genetic support for drug targets was the highest for low-density lipoprotein (LDL)

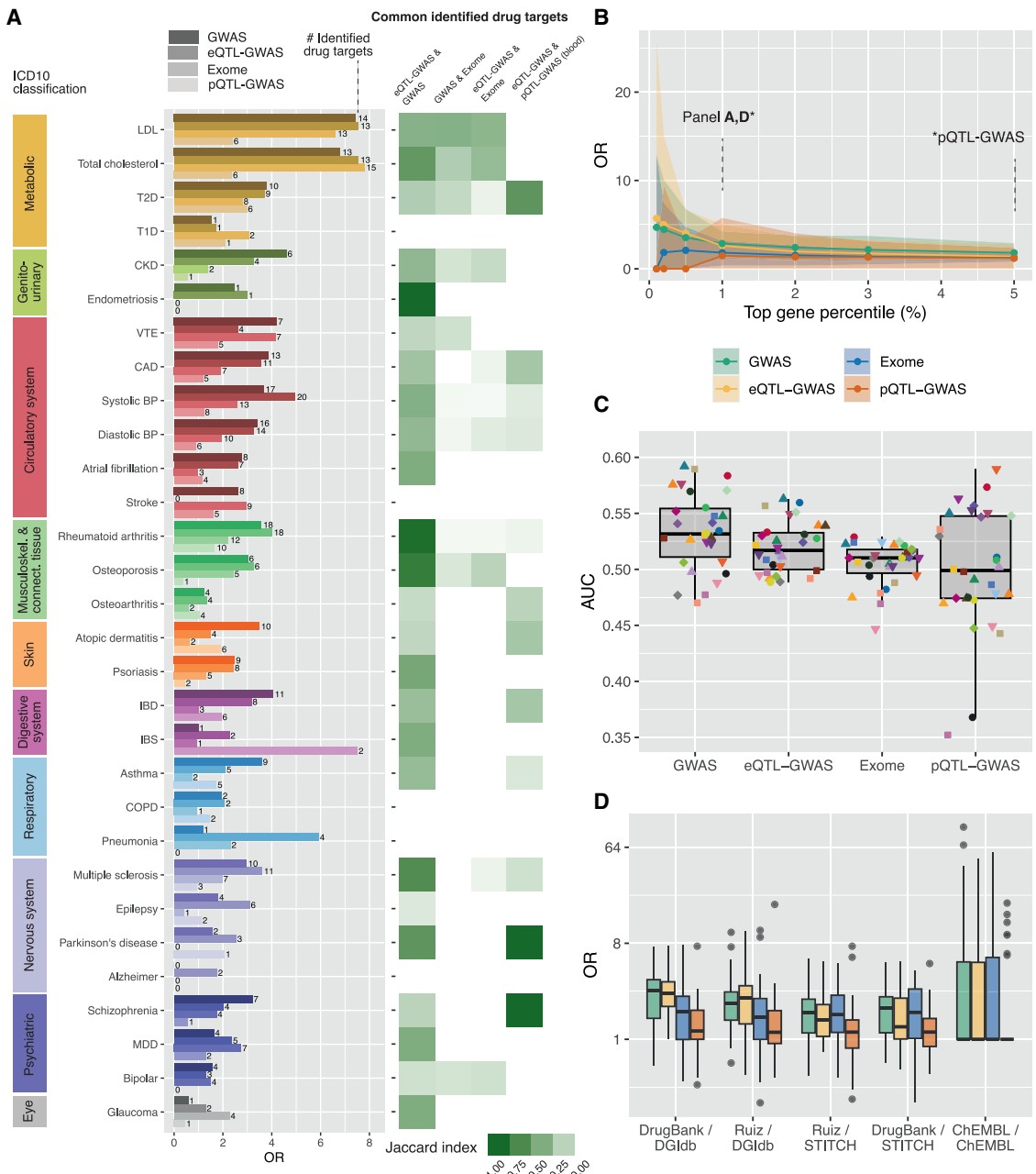

**Figure 3. Enrichment of prioritized genes for drug targets**

(A) Left: bar plot with odds ratios (ORs) calculated from Fisher's exact tests between drug target genes and the top 1% (5% for pQTL-GWAS) prioritized genes for the four tested methods and 30 traits, classified according to trait category. Drug target genes were defined by DrugBank/DGIdb, and only drug target genes that could be tested by the respective method were considered. The number on the right of each bar indicates the number of identified drug target genes. Right: overlap of identified drug target genes between pairs of methods quantified through the Jaccard index. The blood-only eQTL-GWAS gene prioritization method was used for the comparison with the pQTL-GWAS method. Plots using UKBB GWASs only are shown in Figure S3.

(B) ORs at different top prioritized gene percentiles for the four methods. The plotted dots correspond to the median OR across the 30 traits, and the shaded area bounds the 10% and 90% percentiles.

(C) Boxplots showing the area under the receiver operating characteristic curve (AUC) values. AUC values were calculated for each trait as indicated by the points (legend in Figure 2) and using the same background genes and drug target definitions as in (A).

(D) ORs calculated for the five drug target definitions and for all four methods (legend in B). The OR was set to 1 for traits with no identified drug target genes. In (C) and (D), the boxplots bound the 25th, 50th (median, center), and 75th quantiles. Whiskers range from minima (Q1 − (1.5 × IQR)) to maxima (Q3 + (1.5 × IQR)) with points outside representing potential outliers.

and total cholesterol (average ORs of 5.99 and 6.12, respectively). Lowest enrichment ratios were obtained for neuropsychiatric traits (average OR of 1.56) and glaucoma (average OR of 1.14). The average OR across traits was 2.48, 2.68, 1.65, and 1.26 for the GWAS, eQTL-GWAS, Exome, and pQTL-GWAS methods, respectively. We explored a range of top disease gene percentiles (0.1%–5%), and the corresponding ORs are shown in Figure 3B. Restricting disease genes to the top 0.1% for all methods increased the average ORs without changing the method ranking, with average ORs of 3.68, 4.02, 2.40, and 1.44 for the GWAS, eQTL-GWAS, Exome, and pQTL-GWAS methods, respectively. We further analyzed whether identified drug targets were the same across methods and found that prioritized drug target genes were similar between GWAS and eQTL-GWAS methods (average Jaccard index of 0.39), were less so between eQTL-GWAS and pQTL-GWAS methods (blood tissues; average Jaccard index of 0.15), and were very different from Exome identified targets (average Jaccard index of 0.06 between GWAS and Exome and between eQTL-GWAS and Exome methods). Average AUC values across traits were 53.4%, 51.9%, 50.5%, and 49.9% for the GWAS, eQTL-GWAS, Exome, and pQTL-GWAS methods (Figure 3C).

While the number of drugs reported per indication was similar across databases (average of 43.9, 41.8, and 40.4 for Ruiz et al., ChEMBL, and DrugBank, respectively), the average number of reported drug targets was much higher for Ruiz/STITCH (285), Ruiz/DGIdb (274.8), DrugBank/DGIdb (263.4), and DrugBank/STITCH (244.2) than for ChEMBL/ChEMBL (24.8; Table S6). We repeated drug target enrichment calculations for all drug database combinations (Figures 3D and S2). The average ORs for the GWAS/eQTL-GWAS methods were 2.48/2.68, 2.80/2.53, 2.18/2.12, 1.78/1.61, and 1.78/1.51 for DrugBank/DGIdb, ChEMBL/ChEMBL, Ruiz/DGIdb, Ruiz/STITCH, and DrugBank/STITCH, respectively. Overall, the variability in ORs across traits was the highest in the ChEMBL database (Figures 3D and S2), likely due to the low average number of reported drug targets, which leads to very high ORs when drug targets figured among the prioritized genes (e.g., for LDL and total cholesterol), but for many traits drug target genes were not among the prioritized genes (e.g., for type 1 diabetes, atopic dermatitis, and inflammatory bowel disease).

Since enrichment results can differ widely across traits and reference databases, we calculated overall enrichment and AUC values across traits and drug databases, including sensitivity analyses on UKBB data only, to match Exome sample sizes and common background genes (Table S8 and Figure S4; STAR Methods). The overall ORs were 2.17 (UKBB, 1.72), 2.04 (UKBB, 1.67), and 1.81 and 1.31 (UKBB, 1.30) for the GWAS, eQTL-GWAS, and Exome and pQTL-GWAS methods, respectively. There were no significant differences between these four methods in terms of enrichment OR ($p_{diff} > 0.05$, including in the sensitivity analyses). Overall AUCs were 54.3% (UKBB, 52.8%), 52.8% (UKBB, 51.4%), and 51.7% and 51.3% (UKBB, 50.6%) for the GWAS, eQTL-GWAS, and Exome and pQTL-GWAS methods, respectively. Judging by the AUC values, GWAS performed significantly better than eQTL-GWAS ($p_{diff} = 3.1e{-}5$) and also when considering only testable eQTL genes ($p_{diff} = 2.9e{-}4$). When excluding eQTLGen from the tissue-wide eQTL-GWAS, the performance of eQTL-GWAS dropped

slightly (AUC of 52.2% compared with 52.8%; $p_{diff} = 0.019$). Significantly higher AUC values were obtained for GWAS compared with Exome on consortium data ($p_{diff} = 2.2e{-}4$), which was no longer the case on UKBB data ($p_{diff} = 0.06$). The difference between eQTL-GWAS and Exome was not significant on either dataset ($p_{diff} = 0.12$ and 0.77 on consortium and UKBB data, respectively). The number of testable genes was much lower for the pQTL-GWAS method ($\sim$1,870 genes). With this set of background genes, GWAS still scored a higher overall AUC (55.1%, $p_{diff} = 2.1e{-}3$). No difference was observed between the pQTL-GWAS and the tissue-wide or whole blood eQTL-GWAS methods ($p_{diff} = 0.66$ and 0.87, respectively).

### Examples of drug target prioritization ranks

In Figure 4, we highlight drug targets and their gene prioritization ranks for a few examples (complete list in Table S9). Major anti-hypercholesterolemic drug targets *PCSK9* (evolocumab, alirocumab), *HMGCR* (statins), and *NPC1L1* (ezetimibe) were top ranked by all methods (except for no pQTLs being available for *HMGCR* and *NPC1L1*; Figure 4A). HCN4, the target of the antiarrhythmic drug dronedarone, was prioritized as a disease gene for atrial fibrillation only through the GWAS method. Although highly expressed in the atrial appendage and left ventricle of the heart, no eQTL was reported for this gene (Figure 4B). Several antiepileptic drugs target SCN1A, which was highly prioritized by the GWAS and eQTL-GWAS methods, with the strongest MR effect found in the nucleus accumbens (basal ganglia) of the brain (Figure 4C). The antiplatelet drug dipyrimadole used in the prevention and treatment of vascular diseases such as stroke and coronary artery disease is listed to target 23 genes of the *PDE* superfamily in ChEMBL. Of these, four (*PDE4D*, *PDE3A*, *PDE3B*, *PDE6B*) were ranked in the top 1% by the exome method for stroke (Figure 4D). None of the other methods prioritized any of these 23 genes. For coronary artery disease, another superfamily member (*PDE5A*) had a low ranking (<2%) by the GWAS and QTL-GWAS methods, supported by solid GWAS and e/pQTL colocalization (Figure 4E).

### Heritability of drug target transcripts and proteins

Previous drug target enrichment analyses have shown that drug target genes are more likely to have lower residual variance intolerance scores (RVISs), i.e., are less tolerant to change.[1] Furthermore, limited overlap between eQTL and GWAS hits has been found, and it has been suggested that GWAS and eQTL genes are under different selective constraints.[41] Hence, under the assumption that drug target genes are more likely to be key (core) GWAS genes, we expected that drug target genes are less likely to harbor QTLs. To test this hypothesis, we assessed whether drug target transcript or protein levels are less amenable to regulation by common genomic variations, which could explain the lower than expected performance of QTL-GWAS approaches.

To this end, we compared the *cis* heritability of drug target genes vs. non-drug target genes that were measured in the respective studies (i.e., also those with no reported e/pQTLs; STAR Methods), where lower heritability would point toward a negative selection.[42] We conducted the analysis per trait and for each of the five drug target gene definitions; however, we could not observe a clear difference between *cis* heritabilities

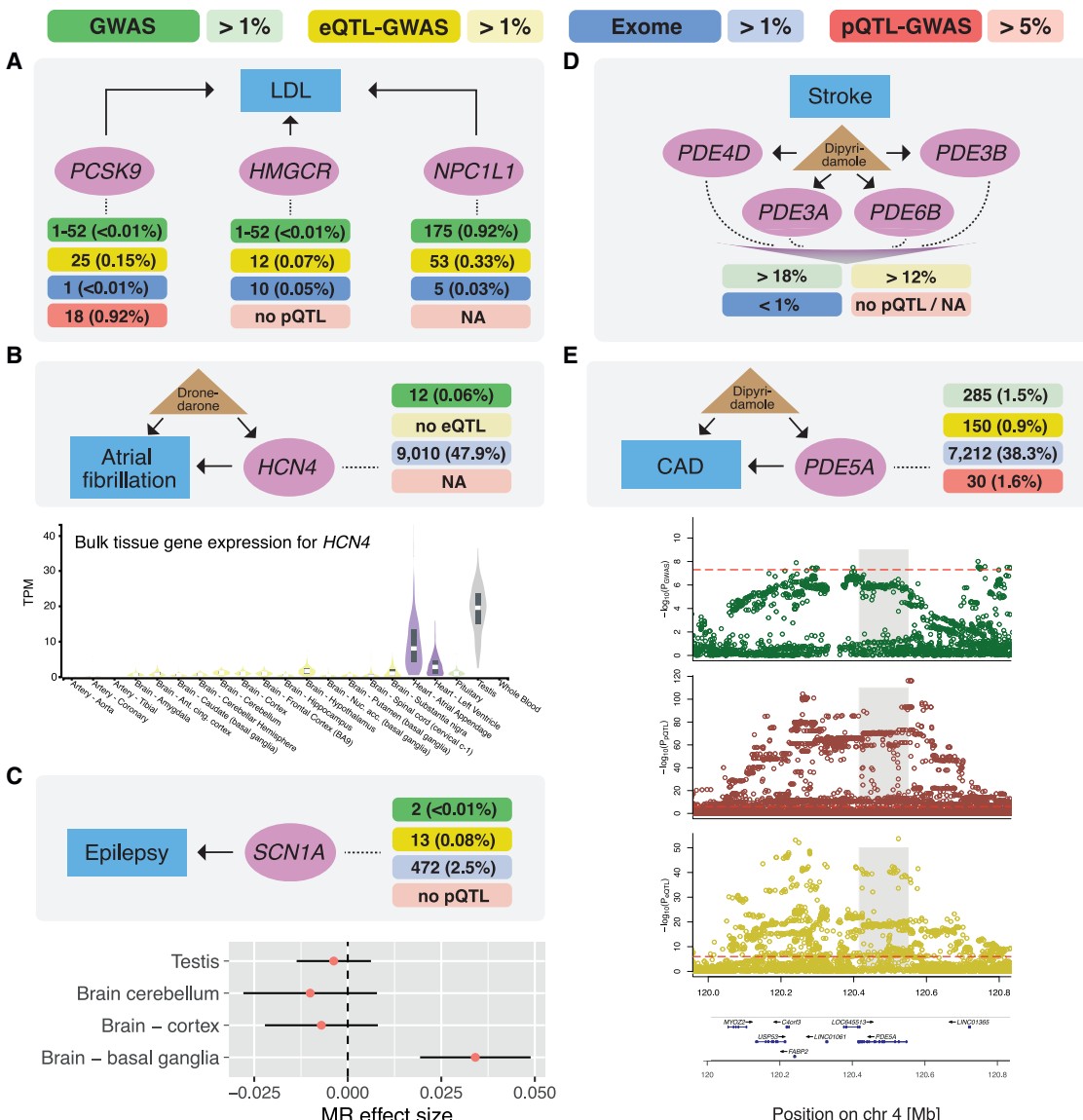

**Figure 4. Examples illustrating drug target genes and their prioritization ranks**

(A) Three drug target genes (*PCSK9* [evolocumab, alirocumab], *HMGCR* [statins], and *NPC1L1* [ezetimibe] shown in purple) for LDL cholesterol (blue box) and their prioritization ranks (top percentiles shown in parentheses) of each of the four methods (GWAS in green, eQTL-GWAS in yellow, Exome in blue, and pQTL-GWAS in red). Genes that were not testable by a given method are reported as NA (no e/pQTL means that the gene was measured, but had no QTL), and a range of ranks (i.e., 1–52) indicates tied p values.

(B) Top plot shows the prioritization ranks of *HCN4*, the target of the antiarrhythmic drug dronedarone. Bottom plot shows the gene expression profile of *HCN4* across GTEx tissues (TPM, transcripts per million) with "testis," "heart-atrial appendage," and "heart-left ventricle" dominating.

(C) Top plot shows the prioritization ranks of *SCN1A* (sodium voltage-gated channel alpha subunit 1), a drug target gene of several antiepileptic drugs. Bottom plot shows Mendelian randomization (MR) effects (red dots) with 95% CI (black bars) across tissues in which there was a significant eQTL.

(D) Antiplatelet drug dipyrimadole and gene prioritization ranks of its multiple drug targets (a non-exhaustive selection) of the phosphodiesterase (*PDE*) superfamily.

(E) Top plot shows the gene prioritization ranks of *PDE5A*, another reported target for dipyrimadole. Bottom plot shows the regional SNP associations ($-\log_{10}(p)$) with coronary artery disease (CAD; GWAS, green), PDE5A protein (pQTL, red), and *PDE5A* transcript (eQTL, yellow) (red dashed lines indicate the significance thresholds of the respective SNP association, and gray shading marks the position of *PDE5A*). Bottom row illustrates the position and strand direction of the genes in the locus.

of drug target and non-drug target genes (Figure S5). While this means that we cannot explain why the QTL-GWAS approach does not perform better, it may also imply that drug target genes are not necessarily typical GWAS genes or so-called core genes.

**Network diffusion to prioritize drug target genes**

Finally, we assessed whether network diffusion can identify drug target genes for which there is no direct genetic evidence. Gene scores from prioritization methods defined the initial distribution

$p_0$ of the diffusion process. This process is regulated by a restart parameter $r$, whereby lower values result in a stronger diffusion (i.e., genes can be prioritized even when distant from initial disease genes; STAR Methods). The stationary distribution was calculated for six different restart parameters, ranging from no diffusion ($r = 1$) to complete diffusion ($r = 0$), and for each of the three networks: the STRING PPI network,[32] an RNA-seq coexpression network (CoXRNAseq),[33] and a coexpression and proteomics network (FAVA).[34] Since the set of testable proteins ($\sim$1,870) is enriched for drug target genes (two-sided binomial test: p = 1.3e−47 for DrugBank/DGIdb; complete results in Table S15; STAR Methods), the AUC values were artificially inflated upon projecting the gene scores onto the network, and pQTL-GWAS results are hence not discussed.

Applying diffusion using the STRING network massively boosted the overlap between the diffused prioritized genes and the drug target genes (Figures 5A, 5B, S6, and S7). At no diffusion, overall AUC values across the 30 traits were 54.3%, 52.8%, and 51.7% for the GWAS, eQTL-GWAS, and Exome methods, respectively, which increased to 68.9%, 67.7%, and 66.9% at a diffusion parameter of $r = 0.6$, and further increased to 73.5%, 72.9%, and 72.3% at stronger diffusion ($r = 0.4$; Figures 5A and S6 and Table S11). A stronger enrichment of prioritized genes for drug targets upon diffusion was also observed when enrichment scores for the top 1% genes were calculated, with overall ORs of 4.63, 5.21, and 5.07 at $r = 0.4$ (Figures 5B and S7 and Table S11). On the other hand, improvements were modest when considering coexpression networks. At $r = 0.6$, overall AUC values increased to 54.9%, 54.7%, and 53.5% in the case of the CoXRNAseq network for the GWAS, eQTL-GWAS, and Exome methods, respectively. Although small, the difference was significant compared with no diffusion ($p_{diff}$ of 5.11e−3, 4.12e−14, and 4.83e−5, respectively). In the same scenario, overall ORs at $r = 0.6$ were 2.28, 2.04, and 1.91, which were not significantly different ($p_{diff} > 0.05$) compared with no diffusion. Likewise, in the FAVA network, overall AUC values at $r = 0.6$ were 55.9%, 54.2%, and 53.6% ($p_{diff}$ compared with no diffusion of 2.23e−5, 3.08e−3, and 7.3e−6), and ORs were 2.38, 2.02, and 1.77 ($p_{diff} > 0.05$), for GWAS, eQTL-GWAS, and Exome methods, respectively (Figures S6 and S7; Tables S11 and S12).

We further assessed which method's AUC values benefited the most from network diffusion. To allow fair comparison with the Exome methods, we used UKBB GWAS data for the GWAS and eQTL-GWAS methods. Across all diffusion parameters $r$, overall AUC values were significantly higher for GWAS compared with eQTL-GWAS in the STRING and FAVA network ($p_{diff} < 4.45e−4$), but not any different in the RNA-seq coexpression (CoXRNAseq) network ($p_{diff} > 0.05$). A nominally significant difference in favor of GWAS compared with Exome was observed only in the STRING network at $r$ values of 0.4, 0.6, and 0.8 ($p_{diff}$ of 0.0262, 7.36e−3, and 0.0146, respectively). No statistical differences were observed between the eQTL-GWAS and the Exome method except for a nominally significant difference in favor of eQTL-GWAS at $r = 0.2$ in the CoXRNAseq network ($p_{diff} = 0.0113$).

When investigating the network connectivity, we observed that drug target genes were significantly more likely to be hub genes, i.e., to have more connections in the network in comparison with other genes (Figures 5C and S8). This observation was particularly strong in the STRING network (mean log-degree = 13.0 vs. 12.3, $p_{diff} = 6.6e−284$ for DrugBank/DGIdb), but also present in the coexpression networks ($\Delta$ log-degree = 0.064, $p_{diff} = 0.011$ for CoXRNAseq; $\Delta$ log-degree = 0.3, $p_{diff} = 6.6e−11$ for FAVA). As a consequence, the network's node degree (a gene's number of connections to other genes adjusted by the edge weight) was found to be a good predictor of drug targets, and the best performance was found for the network degree in STRING (overall AUC = 77.6%, overall OR = 8.71). Given this bias, we generated random initial disease gene scores and determined to what extent genetically informed $p_0$ distributions performed better compared with random $p_0$ distributions. Although the GWAS, eQTL-GWAS, and Exome methods had significantly higher AUC values compared with random score distributions for any given $r$ value in the STRING network ($p_{diff} < 1.62e−7$; Table S12), the performance of a mildly diffused ($r = 0.8$) random score (which is unaware of the target disease) performed significantly better than any disease gene prioritization method without diffusion ($p_{diff}$ of 4.18e−6, 3.58e−10, and 2.10e−12 compared with GWAS, eQTL-GWAS, and Exome, respectively). In line with this observation, the network degree was still significantly better than gene prioritization methods at a stronger diffusion of $r = 0.2$ ($p_{diff}$ of 8.98e−6, 9.87e−13, and 1.89e−11 compared with GWAS, eQTL-GWAS, and Exome, respectively).

### Examples of prioritized genes through network diffusion

In the following, we describe several examples for which drug targets figured among the top 1% genes only after network diffusion (complete list in Table S13). Amyloid-beta precursor protein (APP) targeted by the monoclonal antibody aducanumab in the treatment of Alzheimer's disease (AD) was ranked 506 (top 2.7%) prior to and 152 (top 0.8%) after diffusion on the STRING network ($r = 0.6$; Figure 6A) based on the eQTL-GWAS method. Prioritization was largely influenced by its interacting neighbor apolipoprotein E (APOE), which was the top 5 ranked gene for AD by the eQTL-GWAS method and among the top 6 genes (tied p values) by the GWAS method. Although rare mutations in APP are a known cause of AD,[43] the Exome method did not highly prioritize this gene (>top 10%), likely because of low statistical power due to the younger and healthier nature of the UKBB cohort. Indeed, APP was among the top 1% for the GWAS method, leveraging the AD consortium data, but did not reach the top 10% when restricting the analysis to the UKBB. Tumor necrosis factor (TNF), a drug target in the treatment of inflammatory diseases such as psoriasis, was ranked 1,558th (top 8%; Exome-psoriasis) prior to and 182nd (top 0.98%; $r = 0.6$) post-propagation in the STRING network (Figure 6B). While initially the drug target F2 (coagulation factor II, thrombin) for venous thromboembolism (VTE) ranked only in the top 2%, it moved up to the top 1% regardless of the network used for diffusion at $r = 0.6$ (top 0.9%, 0.6%, and 0.7% for STRING, CoXRNAseq, and FAVA, respectively). In the STRING and CoXRNAseq networks, this boost could largely be attributed to the interacting fibrinogen genes (FGA, FGB, and FGG) that ranked in the top 0.06% (Figure 6C).

**Figure 5. Effect of network diffusion to prioritize drug target genes**

(A) Boxplots showing the area under the receiver operating characteristic curve (AUC) values for each network type (STRING, CoXRNAseq, and FAVA) and method at different restart parameter values $r$. AUC values were calculated for each of the 30 traits, and drug target genes were defined by DrugBank/DGIdb. At an $r$ value of 1 (no network diffusion), the analysis corresponds to the results in Figure 3B, and at an $r$ value of 0, the gene prioritization rank is based simply on the degree of the network nodes. At $r < 1$, the background genes are the genes reported in the respective network. The star next to the pQTL-GWAS method signals that the set of testable genes for this method is enriched for drug target genes, and therefore, higher AUC values were obtained when adding background genes with zero-valued initial scores.

(B) Odds ratios (ORs) between prioritized genes (top 1%) and drug target genes for each network type and method at different $r$ values across the 30 traits (same drug target and background genes as in A). The OR was set to 1 for traits with no identified drug target genes.

(C) Histograms showing the degree distribution of drug target genes and non-drug target genes in each network. The difference in log-degree ($\Delta$) and the p values from two-sided t tests are shown at the top. In (A) and (B), the boxplots bound the 25th, 50th (median, center), and 75th quantile. Whiskers range from minima (Q1 − (1.5 × IQR)) to maxima (Q3 + (1.5 × IQR)) with points above or below representing potential outliers.

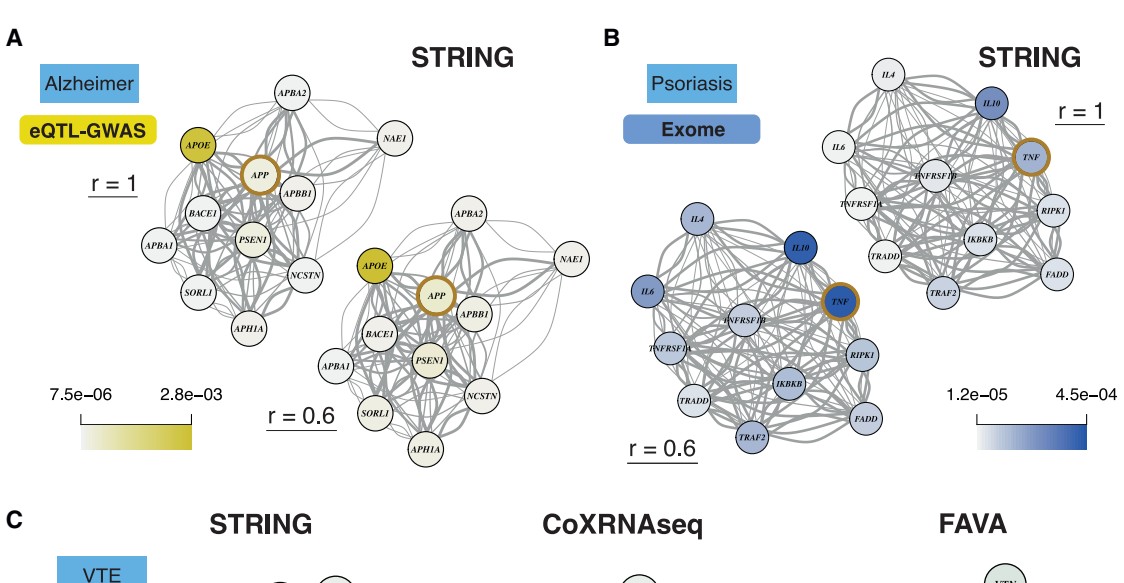

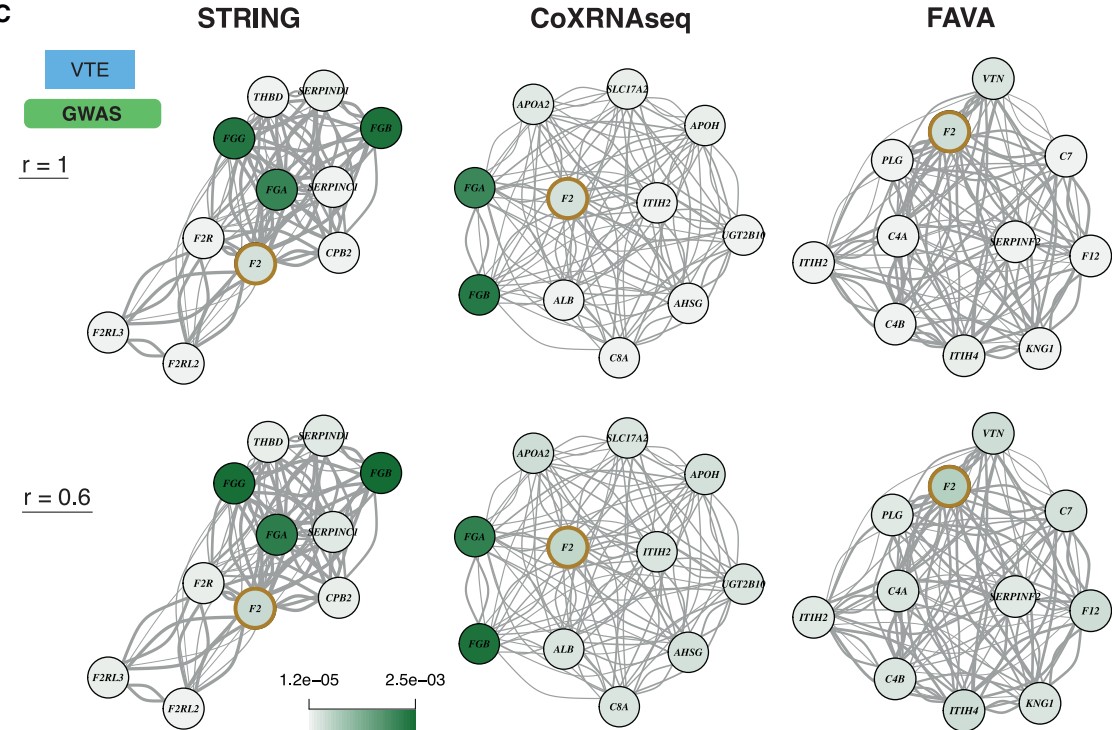

**Figure 6. Examples illustrating prioritized drug target genes through network diffusion**

(A) Top 10 network neighbors of drug target APP (brown circle) and their prioritization values (i.e., normalized node probabilities) by the eQTL-GWAS method for Alzheimer's disease are shown before ($r = 1$) and after diffusion ($r = 0.6$) on the STRING network.

(B) Same representation as in (A) showing Exome prioritization values for psoriasis and tumor necrosis factor (TNF) drug target.

(C) Top 10 network neighbors of drug target F2 (coagulation factor II, thrombin) in the STRING, CoXRNAseq, and FAVA networks. GWAS prioritization values for venous thromboembolism (VTE) are shown before ($r = 1$) and after diffusion ($r = 0.6$) on each network. In each network example (A–C), the drug target gene was among the top 1% prioritized genes only after diffusion at $r = 0.6$.

## DISCUSSION

### Summary of findings

We conducted a comprehensive benchmarking between different genetically informed approaches (GWAS, QTL-GWAS, and Exome) combined with network diffusion to prioritize drug target genes. The strength of our analysis lies in the side-by-side comparison of gene prioritization methods that individually have proven to be successful in identifying drug targets. In line with previous reports, we find a 1.3- to 2.2-fold enrichment for drug targets among (the top 1%) prioritized genes.[1,2] Recently, methods have emerged that combine multiple genetic predictors to derive an aggregate score, often using machine-learning techniques.[27,44,45] These scores have demonstrated high enrichment for drug targets but reveal little about underlying molecular mechanisms. Our aim was to disentangle the importance of

the choice of the ground truth (i.e., drug target genes) and the input data (such as mQTLs, WES) in combination with different molecular networks to highlight added benefits while also exposing weaknesses compared with using GWAS data alone.

### Comparison of gene prioritization methods

Adjusting for differences in background genes and data origins, GWAS yielded higher AUC than eQTL- and pQTL-GWAS, but no significant difference was found with Exome. Genes prioritized by the Exome method were different from those identified by the GWAS and QTL-GWAS methods, which was also reflected in the identified drug targets. While this could imply that rare and common variant genetic architectures are complementary, differences could also be due to power issues. Possibly, with increased sample size, the implicated genes will converge, but the extent to which they can be perturbed by regulatory vs. rare coding variants might remain different. Considering ORs, we lacked the statistical power to claim significant differences between methods, since the number of drug targets among the top 1% prioritized genes can be very low. Overall enrichment ORs for drug targets were 2.17, 2.04, 1.81, and 1.31 for the GWAS, eQTL-GWAS, Exome, and pQTL-GWAS methods, respectively. Although ORs for the pQTL-GWAS method may seem lower, it should be noted that testable proteins (i.e., proteins with pQTLs) accounted for ~10% of GWAS-testable genes. On the same background genes, ORs for the tissue-wide and blood-only eQTL-GWAS methods were 1.38 and 1.22, respectively. For the AUC metric, no significant difference between eQTL-GWAS and pQTL-GWAS was found. In the method comparisons, we considered multiple drug target gene definitions. The number of targets per drug drastically differed between ChEMBL and the DGIdb or STITCH database due to differences in their construct. Drug target genes in the ChEMBL database are manually curated and should not contain false positives, but it remains debatable whether one should consider only primary or also secondary target genes. For instance, ChEMBL lists only HMGCR as a drug target for statins, whereas the DGIdb database also includes APOA5, APOB, and APOE, among others. For this reason, we considered different databases and present enrichment results for both broad and narrow drug target definitions, as well as aggregates.

### Benefits and pitfalls of network diffusion

Network diffusion was beneficial for prioritizing drug target genes with weaker genetic support. A remarkable increase in drug target identification was achieved when using the STRING PPI network. However, this improvement may be due to a circularity in the data generation process, whereby drug target genes are more researched and hence have more chance to be found to interact with other proteins, i.e., they tend to look more hub-like. Although genetically informed gene sets performed better than random ones, the genes prioritized by their node degree in the STRING network resulted in the highest AUC values overall. Thus, care has to be taken when relying on literature-derived gene-gene interactions, as aggressive diffusion will point to the same drug target genes, irrespective of the disease, due to the intrinsic bias stemming from under- and overstudied proteins. While the STRING network resource remains of immense value to identify

interacting proteins, non-random missing of network edges leads to a biased network structure, which makes this resource less suitable as input for discovering new drug targets. The improvements made with coexpression networks, which do not suffer from publication/curation biases, were minor in comparison. Although significant with the AUC metric, ORs were not significantly increased with a diffusion of $r = 0.6$ compared with no diffusion for any of the methods.

### Limitations of the study

Several limitations should be considered. First, we do not take into account the directionality of therapeutic and genetic effects, i.e., whether the drug is an agonist or antagonist. Although found to be less performant than GWAS, QTL-GWAS methods have the advantage of specifying directionality, as opposed to gene scores from the GWAS approach, which ignores SNP effect directions. Second, the mQTL datasets used cover only a small fraction of possible intermediate traits through which SNPs exert their disease-inducing effects.[46] Third, we focus only on common genetic variants when associating transcript and protein levels. With the advent of coupled rare variant-protein level data, either from populations enriched for rare variants or sequencing data,[14,47] more powerful QTL-GWAS methods are likely to emerge that combine mechanistic insights gained from QTL approaches while capturing rare variant associations previously missed. Fourth, drug target data are sparse, which limits the statistical power in benchmarking analyses. Given the required resources to test a drug target in clinical settings, focusing on top ranking genes is of most interest. This scenario is best described with a threshold that defines highly prioritized genes for enrichment analyses. However, ROC curves that quantify the performance at all prioritization thresholds (i.e., use all data at hand) were better powered to detect subtle differences between methods. Resulting AUC values are relatively low (51%–54%), which may be because ranks of genes with non-significant p values are likely unreliable, but these dominate most of the ROC curve. Related to this, even for low false positive rates, there is room for improvement of the gene prioritization methods. Combining prioritization methods could increase AUC values, as suggested by the distinct drug target sets identified by GWAS and Exome methods, as could the integration of additional functional genomic annotations.[27,44] Finally, our analysis compares methods using historical drug discovery data as the ground truth. These data are highly biased, with G-protein-coupled receptors being targets of a third of FDA-approved drugs.[48] Many other genes may be effective targets, but have never been tested in clinical trials. Thus, our results may not reflect how well the tested genetic approaches uncover true disease genes, but rather how well they identify targets that were historically prioritized in drug development processes. Since the emergence of robust GWASs, more and more clinical trials are motivated by genetically informed targets. Thus, drug target databases will tend to overlap better with GWAS-inspired genes, leading to artificially higher overlap.

### Conclusion

To conclude, we systematically evaluated major gene prioritization approaches for their ability to identify approved drug target

genes. Our analyses highlight the power of harnessing multiple data sources by capitalizing on QTLs for mechanistic insights, sequencing data for rare variant associations, GWASs when mQTL signals are missing, and network propagation to leverage gene-gene interactions.

## STAR★METHODS

Detailed methods are provided in the online version of this paper and include the following:

- KEY RESOURCES TABLE
- RESOURCE AVAILABILITY
    - Lead contact
    - Materials availability
    - Data and code availability
- METHOD DETAILS
    - GWAS data
    - GWAS gene scores
    - Molecular QTL-GWAS gene scores
    - Exome gene scores
    - Drug target genes
    - Transcript and protein level heritabilities
    - Networks
    - Network diffusion
- QUANTIFICATION AND STATISTICAL ANALYSIS
    - Concordance of gene scoring methods
    - Drug target enrichment and AUC calculations
    - Enrichment of proteins for drug targets

## SUPPLEMENTAL INFORMATION

## ACKNOWLEDGMENTS

This work was supported by the Swiss National Science Foundation (310030_189147) to Z.K. This research was conducted using the UK Biobank Resource under application 16389. LD was calculated based on the UK10K data resource (EGAD00001000740, EGAD00001000741). Computations were performed on the JURA cluster at the University of Lausanne. We also would like to acknowledge the participants and investigators of the UK Biobank and FinnGen study. We thank Daniel Krefl for his help and support in implementing the PascalX software, Doug Speed for his help and support in calculating heritability estimates, and Liza Darrous for critically reading the draft.

## AUTHOR CONTRIBUTIONS

M.C.S. and Z.K. conceived and designed the study. M.C.S. performed statistical analyses. P.D. provided guidance on statistical analyses. Z.K. supervised all statistical analyses. C.A. contributed to the collection and interpretation of pharmacological and biological data. All the authors contributed by providing advice on interpretation of results. M.C.S. and Z.K. drafted the manuscript. All authors read, approved, and provided feedback on the final manuscript.

## DECLARATION OF INTERESTS

The authors declare no competing interests.

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

# STAR★METHODS

## KEY RESOURCES TABLE

| REAGENT or RESOURCE | SOURCE | IDENTIFIER |
|---|---|---|
| **Deposited data** | | |
| UK Biobank | UK Biobank | https://www.ukbiobank.ac.uk/ |
| UK Biobank GWAS summary statistics | UK Biobank | http://www.nealelab.is/uk-biobank |
| UK Biobank GWAS summary statistics | UK Biobank | https://pan.ukbb.broadinstitute.org |
| FinnGen GWAS summary statistics | FinnGen | https://www.finngen.fi/en/access_results |
| Consortia GWAS summary statistics | Various sources | Table S1 |
| Multiple sclerosis GWAS summary statistics | IMSGC | https://imsgc.net/?page_id=31 |
| Whole blood expression QTLs | eQTLGen | https://www.eqtlgen.org/cis-eqtls.html |
| Tissue-wide expression QTLs | GTEx project | https://gtexportal.org/home/datasets |
| Plasma protein QTLs | deCODE study | https://www.decode.com/summarydata/ |
| Whole exome gene burden tests | GWAS Catalog | accession IDs are in Table S2 |
| UK10K data | UK10K | https://www.uk10k.org/data_access.html |
| DrugBank database | DrugBank | https://go.drugbank.com |
| ChEMBL database | ChEMBL | https://www.ebi.ac.uk/chembl/ |
| DGIdb database | DGIdb | https://www.dgidb.org |
| STITCH database | STITCH | http://stitch.embl.de |
| Ruiz et al. Drug-disease links | [36] | https://doi.org/10.1038/s41467-021-21770-8 |
| STRING network | STRING | https://string-db.org |
| Co-expression network | [33] | https://github.com/molgenis/systemsgenetics/wiki/Downstreamer |
| FAVA network | [34] | https://doi.org/10.5281/zenodo.6803472 |
| **Software and algorithms** | | |
| Main pipeline and analysis code | This paper | https://doi.org/10.5281/zenodo.7857973 |
| PascalX | [49] | https://github.com/BergmannLab/PascalX, |
| SMR-IVW | [50] | https://github.com/masadler/smrivw, |
| METAL | [51] | https://github.com/statgen/METAL |
| R package igraph v1.3.5 | [52] | https://igraph.org |
| R package pROC v1.15.3 | [53] | https://doi.org/10.1186/1471-2105-12-77 |
| biomaRt v2.50.3 | [54] | https://doi.org/10.18129/B9.bioc.biomaRt |
| LDAK software v5.2 | [55] | https://dougspeed.com |

## RESOURCE AVAILABILITY

### Lead contact
Further information and requests for resources should be directed to and will be fulfilled by the lead contact, Zoltán Kutalik (zoltan.kutalik@unil.ch).

### Materials availability
This study did not generate new unique reagents.

### Data and code availability
- This paper analyzes existing, publicly available data. Accession numbers for the datasets are listed in the key resources table.
- Drug target genes and prioritized ranks are included in the supplemental material of this paper.
- All original code has been deposited at Github (https://github.com/masadler/DrugTargetMethodComparison) and archived at Zenodo (https://doi.org/10.5281/zenodo.7857973 ).[56]
- Any additional information required to reanalyze the data reported in this paper is available from the lead contact upon request.

**Cell Genomics**
**Article**

## METHOD DETAILS

### GWAS data

We used the largest (to-date), publicly available GWAS summary statistics for each analyzed condition (Table S1). GWAS data came mostly from consortia specific to the respective disease, and were often a meta-analysis comprising the UKBB. Twenty-four out of the 30 conditions were case/control studies, the remaining 6 being continuous traits: diastolic and systolic blood pressure (DBP and SBP, respectively[57]), low-density lipoprotein and total cholesterol (LDL and TC, respectively[58]), estimated glomerular filtration rate (eGFR[59]) and heel bone mineral density ([58]) proxying chronic kidney disease (CKD) and osteoporosis, respectively. For four traits with low case count in the UK Biobank (< 20,000; chronic obstructive pulmonary disease (COPD), endometriosis, pneumonia and psoriasis) and no large-scale GWAS meta-analysis available, we performed a meta-analysis between the UK Biobank[58] and FinnGen[60] using METAL.[51]

### GWAS gene scores

We used PascalX[9,49] to compute gene scores based on GWAS summary statistics. The software takes as input GWAS p values, gene annotations and LD structure. SNPs are assigned to genes and their squared z-scores are summed. This sum, under the null, was shown to follow a weighted chi-square distribution with weights being defined by the local LD structure from which gene p values can be derived.[9] We applied PascalX with default parameters (gene $\pm$ 50 kB) on protein-coding genes using the Ensembl identifiers and annotations (Ensembl GRCh37.p13 version) and the UK10K reference panel.[61] Across traits, ~19,150 protein-coding genes could be tested which were ranked by their PascalX p value.

### Molecular QTL-GWAS gene scores

We integrated molecular quantitative trait loci (QTL) and GWAS summary statistics using Mendelian randomization (MR) implemented in the smr-ivw software.[22,62,50] The exposure (transcript or protein levels) and outcome disease were instrumented with independent genetic variants, also called instrumental variables (IVs; $r^2 < 0.01$) and used to calculate putative causal effect estimates of the exposure on the outcome ($\beta_{MR}$). IVs were required to be strongly associated to the exposure ($P_{QTL} < 1e-6$) and had to pass the Steiger filter ensuring no significantly stronger effect on the outcome than on the exposure.[63] We used expression QTLs (eQTLs) from the eQTLGen consortium[13] (whole blood; $n = 31,684$) and tissue-specific QTLs from the GTEx v8 release[40] (European ancestry; $n = 65$–573 for 48 tissue types; Table S3) to estimate causal transcript-trait effects. In the eQTLGen dataset there were ~12,550 protein-coding genes with at least 1 IV which increased to ~16,250 when integrating the GTEx dataset. MR results from both datasets (whole blood from eQTLGen and 49 tissues from GTEx) were aggregated by considering the MR causal effect with the lowest p value across tissues (Tables S3 and S4). Protein QTLs (pQTLs) from the deCODE study[14] (whole blood; $n = 35,559$) were used to estimate protein-trait causal effects with ~1,870 proteins having at least 1 IV. Prior to the analysis, e/pQTL and GWAS data were harmonized, palindromic SNPs were removed as well as SNPs with an allele frequency difference > 0.05 between datasets. All transcripts and proteins were mapped to Ensembl identifiers as provided by eQTLGen, GTEx and deCODE.

### Exome gene scores

We used gene burden test results computed on WES data from the UK Biobank.[25] We extracted gene-trait associations based on putative loss of function (pLOF) and deleterious missense variants with MAF < 1% (M3.1 nomenclature in original publication) with phenotypes matching the investigated conditions as indicated in Table S2. Associations were provided for ~18,800 genes which were ranked by the association p value and retrieved by the provided Ensembl identifier.

### Drug target genes

We extracted drug target genes from public resources by combining drug-indication and drug-target links from various databases. A given disease/indication was linked to a drug if the drug was indicated to be prescribed for the selected indication and subsequently, the target genes of these drugs were extracted. For drug-indication pairs we consulted DrugBank, Ruiz et al. and ChEMBL:

- DrugBank 5.0[35] (download: May 2022): DrugBank indications are manually curated from drug labels and underwent an expert review process. Drug indications have their own DrugBank condition numbers and drugs their DrugBank identifiers.

  - Ruiz et al.[36]: A drug-disease dataset was created by querying multiple sources such as the Drug Repurposing Database, the Drug Repurposing Hub, and the Drug Indication Database and extracting information from drug labels, DrugBank and the American Association of Clinical Trials Database. Drug–disease pairs were filtered for FDA-approved treatment relationships. This dataset uses NLM UMLS CUIDS identifiers (National Library of Medicine - Unified Medical Language System Controlled Unique Identifier) for diseases and DrugBank identifiers for drugs.
  - ChEMBL[37] (download: May 2022): ChEMBL drug indications are extracted from multiple sources including DailyMed package inserts, Anatomical Therapeutic Chemical (ATC) classification and ClinicalTrials.gov. Mapping of disease terms to Medical Subject Headings (MeSH) vocabulary and the Experimental Factor Ontology (EFO) is done through a combination of text-mining, automated mapping and manual curation/validation. Drugs are reported with ChEMBL identifiers.

The mapping of GWAS traits to the drug indication identifiers of the respective database is shown in Table S5. Drug target genes were extracted from the DGIdb, STITCH and ChEMBL databases:

- Drug Gene Interaction database (DGIdb) 4.0[38] (release: January 2021): Aggregated drug-gene interactions from multiple sources including DrugBank, Drug Target Commons, the Therapeutic Target Database and Guide to Pharmacology. Genes were matched to Ensembl identifiers using the provided gene vocabulary file. Drugs were reported through DrugBank or ChEMBL identifiers, and mapping from ChEMBL to DrugBank identifiers was done with UniChem,[64] using PubChem IDs as intermediates..
- Search Tool for Interacting CHemicals (STITCH) 5.0[39]: Aggregated drug-protein interaction data from high-throughput experiments data, manually curated datasets and prediction methods. Only high confidence drug-protein relationships (confidence score $\geq$ 700) of the type "inhibition" and "activation" were considered. STITCH uses PubChem Chemical Identifiers (CID) for drugs and mapping to DrugBank IDs was done through the *chemical sources* file provided by STITCH. Protein Ensembl identifiers were mapped to gene Ensembl identifiers using biomaRt (GRCh37, v2.50.3)[54].
- ChEMBL[37] (download: May 2022): ChEMBL provides drug targets which have been manually curated from literature. Drug targets are identified by ChEMBL IDs with mapping to UniProt Accessions provided by ChEMBL. UniProt identifiers were then mapped to gene Ensembl identifiers through the UniProt REST API[65].

In this analysis we considered drug target genes resulting from the following combinations: DrugBank/DGIdb, DrugBank/STITCH, Ruiz/DGIdb, Ruiz/STITCH, and ChEMBL/ChEMBL. The number of drugs and drug target genes per indication is shown in Table S6.

### Transcript and protein level heritabilities

Transcript and protein level *cis*-heritabilities were estimated from QTL effects using a restricted maximum likelihood method (reml) with the LDAK-thin heritability model. The LDAK heritability model assumes that the expected heritability contributed by each SNP depends on its MAF and LD. The analysis was conducted with the LDAK software (v5.2; reml method[55]) based on all SNPs in proximity of the transcript/protein ($\pm$ 500 kB) and the UK10K reference panel.[61] We set the –power to $-0.25$ and the –ignore-weights flag to YES to specify the LDAK-thin heritability model. The analysis was restricted to high-quality SNPs which were defined as being non-ambiguous, having a sample size > 5,000 and a MAF $\geq$ 0.01.

Protein heritabilities were based on the deCODE plasma protein dataset[14] and transcript heritabilities for whole blood on the eQTL-Gen dataset.[13] Of the 14,022 protein-coding transcripts in eQTLGen, reml converged for 12,218. Likewise, 3,716 of the 4,502 autosomal proteins in deCODE converged (estimated *cis*-heritabilities are in Table S10). Genes not converging were omitted in *cis*-heritability downstream analyses.

To calculate the difference in heritabilities between drug target and non-drug target genes, we considered all transcripts and proteins measured in the respective study which were classified accordingly. Per trait, the difference in heritability was then calculated through a two-sided t-test. Heritability tests were only performed for traits with at least three drug targets within the respective set of measured transcripts/proteins.

### Networks

To calculate network diffusion scores, we used the following three networks:

- Search Tool for Retrieval of Interacting Genes/Proteins (STRING) v11[32]: The protein-protein (PPI) interaction network results from predictions based on genomic context information, coexpression, text-mining, experimental biochemical/genetic data and curated databases (curated pathways and protein-complex knowledge). Protein Ensembl identifiers were mapped to gene Ensembl identifiers using biomaRt (GRCh37, v2.50.3).[54] We use interaction confidence scores as edge weights.
- CoXRNAseq[33]: This network was constructed by first performing a principal component analysis on the gene coexpression correlation matrix of 31,499 RNA-seq samples. Reliable principal components were retained from which the final network was constructed via Pearson correlations. We filtered pairwise interactions to only retain those with z-scores above 4. Genes were reported with Ensembl identifiers and z-scores were used as edge weights.
- Functional Associations using Variational Autoencoders (FAVA)[34]: This network is based on single cell RNA-seq read-count data from the Human Protein Atlas and proteomics data from the PRoteomics IDEntifications (PRIDE) database. First, the high-dimensional expression data was reduced into a latent space using variational autoencoders. From this latent space, the network was derived via pairwise Pearson correlations. Each reported interaction has a score which we use as edge weight (final network reports interactions with scores above 0.15). Protein Ensembl identifiers were mapped to gene Ensembl identifiers using biomaRt.

A summary of network properties is given in Table S14. In all analyses, we use weighted networks, and we refer to weighted node degrees (i.e., sum of edge weights linking the node of interest to adjacent nodes) as node degrees.

### Network diffusion

We calculated network diffusion scores based on Markov random walks. Starting from an initial node distribution $\boldsymbol{p}_0$, a stationary distribution is calculated based on network connectivity. This diffusion process depends on a restart parameter $r$ which determines how often the random walker returns to the initial values. Analytically, the stationary distribution ($\boldsymbol{p}_\infty$) is given by:

$$\boldsymbol{p}_\infty = (\boldsymbol{I} - (1 - r)\cdot\boldsymbol{W})^{-1}\cdot\boldsymbol{p}_0 \qquad \text{(Equation 1)}$$

where $\boldsymbol{W}$ is the column-normalized weighted adjacency matrix and $\boldsymbol{I}$ the identity matrix of the same dimension as $\boldsymbol{W}$[66]. The initial node distribution $\boldsymbol{p}_0$ was determined by the squared z-scores derived from the gene p values (normalized to sum up to 1). Genes that could not be tested by a given method had their initial value set to 0. Additionally, we tested the performance of network diffusion on random initial distributions $\boldsymbol{p}_0$. For each trait, a random distribution was generated which all were different, but consistent across analyses. Resulting network diffusion scores $\boldsymbol{p}_\infty$ were ranked for AUC calculations, and the top 1% scored genes were used in the enrichment analyses.

Network manipulations, visualization and degree calculations were performed with the R igraph package v1.3.5.[52]

## QUANTIFICATION AND STATISTICAL ANALYSIS

### Concordance of gene scoring methods

We tested whether prioritized genes were similar or dissimilar between pairs of methods. First, only genes (based on Ensembl identifiers) that were common between the two tested methods were selected into the gene background. Then, prioritized genes were defined at different top percentile cut-offs (0.1%, 0.2%, 0.5%, 1%, 2%, 3%, 5%, 7.5%, 10%). The enrichment of prioritized genes between methods was quantified by a Fisher's exact test using common genes as background genes. When calculating median ORs, ORs of traits for which no prioritized genes overlapped at a given percentile were set to 1. Results of this analysis are presented in the result section "concordance of prioritized genes among gene scoring methods".

### Drug target enrichment and AUC calculations

Enrichment for drug target genes was calculated through two-sided Fisher's exact tests. A contingency table was constructed based on testable genes (i.e., background genes), with genes categorized into prioritized (top 1% or 5% for the pQTL-GWAS) and drug target genes. In rare instances (i.e., pQTL-GWAS background genes and ChEMBL/ChEMBL drug targets) where diagonal values were 0, these were changed to 1. If no prioritized gene coincided with a drug target gene, the resulting OR was set to 1 (for visualization purposes this was not done in barplots where each trait was shown individually). AUC values and standard errors were calculated using the R package pROC v1.15.3.[53]

Log-OR and AUC values (both are denoted $b_i$ herein) were aggregated across traits and drug databases ($m = 30\cdot5 = 150$ observations per method) as follows:

$$\overline{b} = \frac{1}{m}\sum_{i}^{m} b_i \qquad \text{(Equation 2)}$$

with corresponding variance:

$$\text{var}(\overline{b}) = 1' \cdot \boldsymbol{S} \cdot \boldsymbol{R} \cdot \boldsymbol{S}\cdot 1 / m^2 \qquad \text{(Equation 3)}$$

where $\boldsymbol{S}$ is a diagonal matrix of size $mxm$ containing standard errors of $b_i$ and $\boldsymbol{R}$ is the correlation matrix between drug databases and traits. This matrix was derived from the Kronecker product of the drug database correlation matrix and phenotypic trait correlation matrix (Tables S6 and S7). The drug database correlation matrix was derived on the gene level (i.e., 1 if the gene was a drug target for any of the 30 traits, 0 if not) and the phenotypic trait correlation on individual-level data from the UKBB (codes in Table S1B). $\overline{b}$ was referred to as the overall AUC/ log-OR (overall OR after an exponential transformation).

To calculate the statistical difference of $\overline{b}_1$ and $\overline{b}_2$ for method 1 and 2, respectively, we derived the variance of the difference as follows:

$$\text{var}(\overline{b}_1 - \overline{b}_2) = \text{var}(\overline{b}_1) + \text{var}(\overline{b}_2) - 2\cdot\text{cov}(\overline{b}_1,\overline{b}_2) \qquad \text{(Equation 4)}$$

with $\text{cov}(\overline{b}_1,\overline{b}_2) \approx r \times (1' \cdot \boldsymbol{S}_1 \cdot \boldsymbol{R} \cdot \boldsymbol{S}_2 \cdot 1 / m^2)$, where $r$ is the empirical correlation between $\boldsymbol{b}_1$ and $\boldsymbol{b}_2$. From the resulting z-score, a two-sided p value was calculated and significance was defined at a p value below 0.05. Results of these analyses are presented in the result sections "enrichment of prioritized genes for drug targets" and "network diffusion to prioritize drug target genes".

### Enrichment of proteins for drug targets

We conducted binomial tests to verify whether the set of testable (i.e., at least 1 pQTL) and measured proteins (~1,870 and ~ 4,450, respectively) were enriched for drug target genes. We performed the analysis on each of the five drug target definitions and proceeded as follows: 1) we extracted the number of testable/measured proteins that are drug targets ("number of successes"), 2) considering all protein-coding autosomal genes (19,430), we extracted those that are drug targets ("number of trials"), 3) we determined the proportion of testable/measured proteins among all protein-coding genes ("expected probability of success"). From these numbers, we conducted two-sided exact binomial tests (Table S15).

