## [Document S2. Transparent peer review records for Sadler et al · Cell Genomics]

Multi-layered genetic approaches to identify approved drug targets

Author list

Marie C. Sadler, Chiara Auwerx, Patrick Deelene, Zoltán Kutalik

Summary

Initial submission: Received : February 27th 2023

Scientific editor: Judith Nicholson

First round of review: Number of reviewers: 2
Revision invited : April 5th 2023
Revision received : April 25th 2023

Second round of review: Number of reviewers: 1
Accepted : 16th May 2023

Data freely available: Yes

Code freely available: Yes

This transparent peer review record is not systematically proofread, type-set, or edited. Special characters, formatting, and equations may fail to render properly. Standard procedural text within the editor's letters has been deleted for the sake of brevity, but all official correspondence specific to the manuscript has been preserved.

Referees' reports, first round of review

Reviewer 1

In this manuscript, Sadler et al. perform an extensive enrichment analysis comparing the genes that are implicated in GWAS and exome-sequencing studies with genes that are the targets of existing therapeutics. This type of analysis has been used as a proxy for the question of whether genetic evidence is useful for finding novel therapeutics, as well. They also perform interesting enrichment analyses involving the neighbors of GWAS- and exome-associated genes in biological networks. Compared with previous analyses along these lines that I am aware of, this study is much more extensive in the amount of data that was used in every stage of the analysis, including different gene-prioritization strategies, multiple drug target gene sets, and multiple biological network datasets.

Overall, the manuscript was interesting and well executed, and I have mostly minor comments. I have one major comment about the section on transcript and protein level heritability.

Major comment

1. The "Heritability and polygenicity of drug target transcripts" subsection feels extraneous, and it uses problematic definitions/estimators of both heritability and polygenicity, and it seems extraneous in the larger context of the paper. This section relates to an interesting phenomenon, discussed in ref. 40 and in Mostafavi et al. bioRxiv, that biologically important genes might be depleted of eQTL effects. This phenomenon is tangentially related to this manuscript in that it may explain the underwhelming performance of the eQTL-GWAS approach, but it is not closely related to the main message of the paper. As it is currently written, the text does not make these connections very clear, and the section feels extraneous.

Moreover, the estimators (definitions?) of heritability and polygenicity in this section are not appropriate. "Heritability" is estimated by adding up the effects of significant effects, with LD clumping. This is not an appropriate estimator because it only includes the effects of significant effects. "Polygenicity" is quantified by counting the number of significant effects, which has the same issue. I am not aware of a good existing approach to robustly quantify polygenicity for cis-eQTL data, and some researchers think polygenicity ought to mean something like "the

number of significant hits," so maybe the definition is acceptable. For "heritability", on the other hand, there exist methods to estimate cis-heritability (e.g. GCTA), and nobody defines it as a property of the genome-wide significant variants.

I think the manuscript would be improved either by extensively re-writing and re-thinking this subsection, or by simply removing it.

Minor comments

2. For the "exome" strategy, a key parameter is the allele frequency threshold. In general, variants with frequencies close to this threshold will dominate the test statistics (this is the opposite behavior as in GWAS, where the threshold is a lower bound). I assume the results would look quite different with a different threshold, and I suggest to perform a sensitivity analysis of the main drug-target-enrichment analysis looking at different allele frequencies, which might help to prioritize genes with larger effect sizes.

3. What should we make of the fact that the STRING PPI network is so much more successful at identifying drug targets compared with genetic approaches? Perhaps this is related to, but distinct from, an observation in Kim et al. 2020 AJHG [?], who showed that genes with high "network centrality" were highly enriched for heritability. I think this finding is striking, and even though it has nothing to do with the genetic data per se, it could use more discussion.

4. The example of APP in Alzheimer's disease is confusing. Rare APP mutations are a known cause of AD, so if this gene is not highly ranked via the exome approach, it suggests an issue with power rather than anything else (not surprising, since UK Biobank skews young). APOE, on the other hand, should be the top GWAS hit by far; why is it only top 5? I don't necessarily think this indicates a problem with the approach, but maybe this example isn't the most reassuring.

Reviewer 2

A primary goal of genetic association studies is the identification of potential drug targets for complex diseases; in this manuscript, the authors present a comprehensive examination of the ability of contemporary methods to successfully identify known drug mechanisms for a broad range of clinical disorders. There are two inter-related major strengths of this manuscript: 1) it addresses a question that lies at the heart of biomedical genetics; and 2) it does so in an exhaustive fashion. Specifically, it examines: 1) a broad range of disorders

(n=30); 2) a range of association strategies (GWAS, eQTL, pQTL, and Exome/rare variant association); 3) a range of chemoinformatic databases for annotation of drug targets; and 4) a range of potential degrees of "distance" between the primary (genetically-identified) target and the drug target, with distance defined in the context of protein interaction networks.

There is only two notable weaknesses in the manuscript:

1) A relatively minor weakness is that the discussion section could be more strongly worded. For example, it appears that an important conclusion of this work is that the STRING dataset is inappropriate for future work in this area, and this should be stated clearly. Similarly, it should be stated clearly that genetic association gives a 1.5-2-fold enrichment for drug targets, consistent with less comprehensive prior reports. At the same time, the AUC is sufficiently low that additional directions for improvement should be suggested.

2) A more significant weakness is that 4 association strategies that are compared are substantially different in terms of available sample size and statistical power at the present time. While the authors acknowledge the limitations specific to the pQTL dataset, it is generally true that available exome studies are less well-powered compared to GWAS. What would the results be if the GWAS were downsampled to match the available exome data for each disease? Similarly, eQTL are differentially powered relative to the tissue of interest that is available; the authors utilize the tissue with the strongest association, but that might advantage some disorders (e.g., those primarily expressed in blood) compared to other disorders expressed in tissues less well represented in available eQTL reference datasets.

Authors' response to the first round of review

Color code:

Answers to the reviewers are written in green.

Unchanged elements borrowed from the manuscript are in blue (*italic*), and changed text quoted from the manuscript is in brown (*italic*).

Reviewer #1

In this manuscript, Sadler et al. perform an extensive enrichment analysis comparing the genes that are implicated in GWAS and exome-sequencing studies with genes that are the targets of existing therapeutics. This type of analysis has been used as a proxy for the question of whether genetic evidence is useful for finding novel therapeutics, as well. They also perform interesting enrichment analyses involving the neighbors of GWAS- and exome-associated genes in biological networks. Compared with previous analyses along these lines that I am aware of, this study is much more extensive in the amount of data that was used in every stage of the analysis, including different gene-prioritization strategies, multiple drug target gene sets, and multiple biological network datasets.

Overall, the manuscript was interesting and well executed, and I have mostly minor comments. I have one major comment about the section on transcript and protein level heritability.

We thank the reviewer for the positive assessment of our work and the constructive feedback. We revised the transcript and protein level heritability section and addressed all remaining comments. Please find our answers below.

Major comment

1. The "Heritability and polygenicity of drug target transcripts" subsection feels extraneous, and it uses problematic definitions/estimators of both heritability and polygenicity, and it seems extraneous in the larger context of the paper. This section relates to an interesting phenomenon, discussed in ref. 40 and in Mostafavi et al. bioRxiv, that biologically important genes might be depleted of eQTL effects. This phenomenon is tangentially related to this manuscript in that it may explain the underwhelming performance of the eQTL-GWAS approach, but it is not closely related to the main message of the paper. As it is currently written, the text does not make these connections very clear, and the section feels extraneous.

Moreover, the estimators (definitions?) of heritability and polygenicity in this section are not appropriate. "Heritability" is estimated by adding up the effects of significant effects, with LD clumping. This is not an appropriate estimator because it only includes the effects of significant effects. "Polygenicity" is quantified by counting the number of significant effects, which has the same issue. I am not aware of a good existing approach to robustly quantify polygenicity for cis-eQTL data, and some researchers think polygenicity ought to mean something like "the number of significant hits," so maybe the definition is acceptable. For "heritability", on the other hand, there exist methods to estimate cis-heritability (e.g. GCTA), and nobody defines it as a property of the genome-wide significant variants.

I think the manuscript would be improved either by extensively re-writing and re-thinking this subsection, or by simply removing it.

We agree with the reviewer that summing up effects of significant QTLs is an approximate method to estimate *cis*-heritability and more appropriate methods such as GCTA exists. We believe that the "Heritability and polygenicity of drug target transcripts and proteins" result

section is of great interest to situate this work in the context of drug target genes having lower RVIS, i.e. being less tolerant to change (Nelson et al. 2015, Nature Genetics, ref. 1) and to relate to previous work that showed limited overlap between eQTL and GWAS hits (Mostafavi et al. bioRxiv) as a consequence of biologically important genes being depleted of QTLs.

We therefore revised this section by omitting polygenicities for which we don't have good quantifications and by calculating *cis*-heritabilities using a restricted maximum likelihood method (reml) with the LDAK-thin heritability model. Unlike the GCTA model which assumes that all SNPs contribute equally, the LDAK-thin model assumes that the heritability contributed by each SNP depends on its minor allele frequency (MAF) and linkage disequilibrium (LD) structure. Heritability calculations were conducted with the LDAK software and the corresponding method section was updated to read as follows:

Method section : Transcript and protein level heritabilities

Transcript and protein level cis-heritabilities were estimated from QTL effects using a restricted maximum likelihood method (reml) with the LDAK-thin heritability model. The LDAK heritability model assumes that the expected heritability contributed by each SNP depends on its MAF and LD. The analysis was conducted with the LDAK software (v5.2; reml method \cite{speed2020evaluating}) based on all SNPs in proximity of the transcript/protein (± 500 kB) and the UK10K reference panel \cite{uk10k2015uk10k}. We set the --power to -0.25 and the --ignore-weights flag to YES to specify the LDAK-thin heritability model. The analysis was restricted to high-quality SNPs which were defined as being non-ambiguous, having a sample size $> 5,000$ and a $MAF \geq 0.01$.

Protein heritabilities were based on the deCODE plasma protein dataset \cite{ferkingstad2021large} and transcript heritabilities for whole blood on the eQTLGen dataset \cite{vosa2021large}. Of the 14,022 protein-coding transcripts in eQTLGen, reml converged for 12,218. Likewise, 3,716 of the 4,502 autosomal proteins in deCODE converged (estimated cis-heritabilities are in Table S11-12). Genes not converging were omitted in cis-heritability downstream analyses.

To calculate the difference in heritabilities between drug target and non-drug target genes, we considered all transcripts and proteins measured in the respective study which were classified accordingly. Per trait, the difference in heritability was then calculated through a two-sided t-test. Heritability tests were only performed for traits with at least three drug targets within the respective set of measured transcripts/proteins.

Since estimating heritabilities from summary statistics requires large sample sizes to obtain meaningful estimates, we did not extend the analysis to eQTLs from the GTEx consortium. Based on the new heritability estimates, we updated Supplementary Figure 5 that shows the results of the "Heritability of drug target transcripts and proteins" section:

Transcripts (whole blood)

Proteins (whole blood)

$-\log_{10}(p\text{-value}) \times \text{sign}(h^2)$

-5.0 -2.5 0.0 2.5 5.0

Supplemental Figure 5. Difference in heritability of drug target compared to non-drug target measured transcript and protein levels. For each trait, the difference in heritability was calculated through a two-sided t-test. When the difference was negative (i.e., drug target genes were less heritable), the $-\log_{10}(p\text{-value})$ is plotted in blue, otherwise in red. Traits for which the difference was nominally significant ($p\text{-value} < 0.05$), are indicated with a star. If less than three drug target genes could be tested for a trait, a grey box is plotted.

Unlike with the previous *cis*-heritability estimates, no consistent trend was observed for drug target genes being less heritable than non-drug target genes. The result section was updated accordingly, and we now also refer to the findings of Mostafavi et al. bioRxiv which showed limited overlap between eQTLs and GWAS hits.

Result section: Heritability of drug target transcripts and proteins

Previous drug target enrichment analyses have shown that drug target genes are more likely to have lower residual variance intolerance scores (RVIS), i.e., are less tolerant to change (cite{nelson2015support}). Furthermore, limited overlap between eQTL and GWAS hits has been found and it has been suggested that GWAS and eQTL genes are under different selective constraints (Mostafavi et al., 2022). Hence, under the assumption that drug target genes are more likely to be key (core) GWAS genes, we expected that drug target genes are less likely to harbor eQTLs. To test this hypothesis, we assessed whether drug target transcript or protein levels are less amenable to regulation by common genomic variations, which could explain the lower than expected performance of QTL-GWAS approaches.

*To this end, we compared the *cis*-heritability of drug target genes versus non-drug target genes that were measured in the respective studies (i.e., also those with no reported e/pQTLs; Methods) where lower heritability would point towards a negative selection (cite{o2019extreme}). We conducted the analysis per trait and for each of the five drug target gene definitions, however, could not observe a clear difference between *cis*-heritabilities of drug target and non-drug target genes (Figure S5). While this means that we cannot explain*

why the QTL-GWAS approach does not perform better, it may also imply that drug target genes are not necessarily typical GWAS genes or so-called core genes.

Since there was no strong support for drug target genes being protected from genomic perturbations, we deleted the corresponding paragraph from the Discussion.

Minor comments

2. For the "exome" strategy, a key parameter is the allele frequency threshold. In general, variants with frequencies close to this threshold will dominate the test statistics (this is the opposite behavior as in GWAS, where the threshold is a lower bound). I assume the results would look quite different with a different threshold, and I suggest to perform a sensitivity analysis of the main drug-target-enrichment analysis looking at different allele frequencies, which might help to prioritize genes with larger effect sizes.

As shown by Backman et al. Nature, 202, the allele frequency can indeed influence the prioritization of genes, however, most associations (64.1%) were shown to be unaffected when changing the MAF threshold from 1% to 0.1% (Extended Figure 6a in Backman et al.) and only 11% of associations became stronger when setting the MAF threshold to 0.1%, whereas 24.9% got weaker. In line, statistical power was shown to be the highest when using a MAF of 1% (Result section "Effect of burden test composition" in Backman et al.). In our analyses, we define the prioritized genes as those with the top 1% evidence of excess burden, which certainly includes many false positive genes (as not all of those survive family-wise error control). Since we wanted the top 1% to contain as many true positives as possible we opted for the best-powered burden results. To this end we chose the 1% MAF filter for all analyses. To further support this choice, we prepared the QQ-plots for the burden test statistics (P-values) for various MAF filters (0.001%, 0.01%, 0.1% and 1%) for LDL cholesterol as outcome. Top 1% genes (which define the prioritized genes in our drug target enrichment analysis) are colored in golden and for these we calculated the median chi-square statistics. As the MAF filter decreases, less burden test associations become significant as indicated by decreasing chi-square statistics and a deviation from the diagonal at higher expected values.

3. What should we make of the fact that the STRING PPI network is so much more successful at identifying drug targets compared with genetic approaches? Perhaps this is related to, but distinct from, an observation in Kim et al. 2020 AJHG [?], who showed that genes with high "network centrality" were highly enriched for heritability. I think this finding is striking, and even though it has nothing to do with the genetic data per se, it could use more discussion.

We updated the discussion to better highlight the bias of the STRING network and its implications in terms of drug target discovery. The concerned section now reads as follows:

*Network diffusion was beneficial for prioritizing drug target genes with weaker genetic support. A remarkable increase in drug target identification was achieved when using the **STRING** protein-protein interaction network. However, this improvement may be due to a circularity in the data generation process, whereby drug target genes are more researched, hence have more chance to be found to interact with other proteins, i.e. tend to look more hub-like. Although genetically-informed gene sets performed better than random ones, the genes prioritized by their node degree in the **STRING** network resulted in the highest AUC values*

overall. Thus, care has to be taken when relying on literature-derived gene-gene interactions, as aggressive diffusion will point to the same drug target genes, irrespective of the disease, due to the intrinsic bias stemming from under- and over-studied. While the STRING network resource remains of immense value to identify interacting proteins, non-random missing of network edges leads to a biased network structure which makes this resource less suitable as input for discovering new drug targets. The improvements made with co-expression networks, which do not suffer from publication/curation biases, were minor in comparison. Although significant with the AUC metric, ORs were not significantly increased with a diffusion of $r = 0.6$ as compared to no diffusion for any of the methods.

Indeed, the observation in Kim et al. AJHG, 2020 that high “network centrality” is enriched for disease heritability is striking. We made a quick analysis correlating network connectivity (i.e. gene node degrees) to the newly calculated transcript *cis*-heritabilities. While no significant correlation was found with the STRING network, significant negative correlations were found with the co-expression networks (p-values of 2.1×10^{-6} and 3.9×10^{-3} for the CoXRNAseq and FAVA networks, respectively). These negative correlations as opposed to the positive enrichments of network centrality with disease heritabilities may be in line with the discrepant architectures of transcript *cis*-heritabilities and GWAS heritabilities as shown by Mostafavi et al., 2022. However, proving this point would need further analyses. Given that we no longer find that drug target genes have lower *cis*-heritabilities, we decided not to include these results. Although, we believe that deeper analyses on heritabilities and network connectivity would be a great addition to our current results, this exceeds the scope of this work.

4. The example of APP in Alzheimer's disease is confusing. Rare APP mutations are a known cause of AD, so if this gene is not highly ranked via the exome approach, it suggests an issue with power rather than anything else (not surprising, since UK Biobank skews young). APOE, on the other hand, should be the top GWAS hit by far; why is it only top 5? I don't necessarily think this indicates a problem with the approach, but maybe this example isn't the most reassuring.

To avoid confusion, we updated the description of this example in the result section. The example only focuses on gene prioritization ranks of the eQTL-GWAS approach, before and after diffusion on the STRING network. Prioritization ranks of the GWAS and Exome method are not considered in this example per se, however, we now include them for completion. As correctly guessed by the reviewer, APP was most likely not picked up by the Exome method because of a bias towards young individuals in the UKBB. In fact, APP was among the top 1% for the GWAS method when using consortium data, but it was not at all prioritized when restricting the analysis to the UKBB data. The paragraph has been updated as follows:

Amyloid-beta precursor protein (APP) targeted by the monoclonal antibody aducanumab in the treatment of Alzheimer's disease (AD) was ranked 506th (top 2.7%) prior and 152nd (top 0.8%) after diffusion on the STRING network ($r = 0.6$; Figure 6a) based on the eQTL-GWAS method. Prioritization was largely influenced by its interacting neighbour Apolipoprotein E (APOE) which was among the top 5 ranked genes for AD by the eQTL-GWAS method and among the top 6 genes (tied p-values) by the GWAS method. Although rare mutations in APP are a known cause for AD (O'Brien and Wong, 2011), the Exome method did not highly

prioritize this gene (> top 10%), likely because of low statistical power due to the younger and healthier nature of the UKBB cohort. Indeed, APP was among the top 1% for the GWAS method leveraging the AD consortium data, but did not reach the top 10% when restricting the analysis to the UKBB.

Figure 6. Examples illustrating network diffusion to prioritize drug target genes. **a** Top ten network neighbours of drug target APP (brown circle) and their prioritization values (i.e., normalized node probabilities) of the eQTL-GWAS method for Alzheimer's disease are shown before ($r = 1$) and after diffusion ($r = 0.6$) on the STRING network.

Reviewer #2

A primary goal of genetic association studies is the identification of potential drug targets for complex diseases; in this manuscript, the authors present a comprehensive examination of the ability of contemporary methods to successfully identify known drug mechanisms for a broad range of clinical disorders. There are two inter-related major strengths of this manuscript: 1) it addresses a question that lies at the heart of biomedical genetics; and 2) it does so in an exhaustive fashion. Specifically, it examines: 1) a broad range of disorders (n=30); 2) a range of association strategies (GWAS, eQTL, pQTL, and Exome/rare variant association); 3) a range of chemoinformatic databases for annotation of drug targets; and 4) a range of potential degrees of "distance" between the primary (genetically-identified) target and the drug target, with distance defined in the context of protein interaction networks.

We thank the reviewer for the positive evaluation of our work and appreciate the comments to strengthen this work. Please find our answers below.

There is only two notable weaknesses in the manuscript:

1) A relatively minor weakness is that the discussion section could be more strongly worded. For example, it appears that an important conclusion of this work is that the STRING dataset is inappropriate for future work in this area, and this should be stated clearly. Similarly, it should be stated clearly that genetic association gives a 1.5-2-fold enrichment for drug targets, consistent with less comprehensive prior reports. At the same time, the AUC is sufficiently low that additional directions for improvement should be suggested.

We updated the discussion 1) to put a stronger emphasis on the limitations of the STRING data, 2) to iterate more explicitly that gene prioritization methods give a 1.3-2.2-fold enrichment for drug target genes and 3) to explain relatively low AUC values and suggest future directions for improvement.

Accordingly, the first paragraph of the discussion now reads as follows:

We conducted a comprehensive benchmarking between different genetically informed approaches (GWAS, QTL-GWAS and Exome) combined with network diffusion to prioritize drug target genes. The strength of our analysis lies in the side-by-side comparison of gene prioritization methods that individually have proven to be successful in identifying drug targets. In line with previous reports, we find a 1.3 to 2.2-fold enrichment for drug targets among (the top 1%) prioritized genes \cite{nelson2015support, king2019drug}. Recently, methods have emerged that combine multiple genetic predictors to derive an aggregate score often using machine-learning techniques \cite{fang2019genetics, mountjoy2021open, forgetta2022effector}. These scores demonstrated high enrichment for drug targets but reveal little about underlying molecular mechanisms. Our aim was to disentangle the importance of the choice of the ground truth (i.e., drug target genes), the input data (such as molecular QTLs, WES) in combination with different molecular networks to highlight added benefits while also exposing weaknesses compared to using GWAS data alone.

The paragraph about the bias in the STRING network was updated as follows:

*Network diffusion was beneficial for prioritizing drug target genes with weaker genetic support. A remarkable increase in drug target identification was achieved when using the **STRING protein-protein interaction** network. However, this improvement may be due to a circularity in the data generation process, whereby drug target genes are more researched, hence have*

more chance to be found to interact with other proteins, i.e. tend to look more hub-like. Although genetically-informed gene sets performed better than random ones, the genes prioritized by their node degree in the STRING network resulted in the highest AUC values overall. Thus, care has to be taken when relying on literature-derived gene-gene interactions, as aggressive diffusion will point to the same drug target genes, irrespective of the disease, due to the intrinsic bias stemming from under- and over-studied. While the STRING network resource remains of immense value to identify interacting proteins, non-random missing of network edges leads to a biased network structure which makes this resource less suitable as input for discovering new drug targets. The improvements made with co-expression networks, which do not suffer from publication/curation biases, were minor in comparison. Although significant with the AUC metric, ORs were not significantly increased with a diffusion of $r = 0.6$ as compared to no diffusion for any of the methods.

We updated the paragraph about limitations to include future directions for improvement:

Several limitations should be considered. First, we do not take into account directionality of therapeutic and genetic effects, i.e., whether the drug is an agonist or antagonist. Although found to be less performant than GWAS, QTL-GWAS methods have the advantage of specifying directionality, as opposed to gene scores from the GWAS approach which ignores SNP effect directions. Second, the used molecular QTL data sets cover only a small fraction of possible intermediate traits through which SNPs exert their disease-inducing effects \cite{yao2020quantifying}. Third, we only focus on common genetic variants when associating transcript and protein levels. With the advent of coupled rare variant-protein level data, either from populations enriched for rare variants or sequencing data \cite{ferkingstad2021large, dhindsa2022influences}, more powerful QTL-GWAS methods are likely to emerge that combine mechanistic insights gained from QTL approaches while capturing rare variant associations previously missed. Fourth, drug target data are sparse which limits the statistical power in benchmarking analyses. Given the required resources to test a drug target in clinical settings, focusing on top ranking genes is of most interest. This scenario is best described with a threshold that defines highly prioritized genes for enrichment analyses. However, ROC curves that quantify the performance at all prioritization thresholds (i.e. use all data at hand) were better powered to detect subtle differences between methods. Resulting AUC values are relatively low (51%-54%), which may be because ranks of genes with non-significant p-values are likely unreliable, but these dominate most of the ROC curve. Related to this, even for low false positive rates there is room for improvement of the gene prioritization methods. Combining prioritization methods could increase AUC values as suggested by the distinct drug target sets identified by GWAS and Exome methods as could the integration of additional functional genomic annotations \cite{fang2019genetics, mountjoy2021open}. Finally, our analysis compares methods using historical drug discovery data as the ground truth. This data is highly biased with G-protein-coupled receptors being targets of a third of FDA-approved drugs \cite{hauser2018pharmacogenomics}. Many other genes may be effective targets, but have never been tested in clinical trials. Thus, our results may not reflect how well the tested genetic approaches uncover true disease genes, but rather how well they identify targets that were historically prioritized in drug development processes. Since the emergence of robust GWAS, more and more clinical trials are motivated by genetically informed targets. Thus, drug target databases will tend to overlap better with GWAS-inspired genes, leading to artificially higher overlap.

2) A more significant weakness is that 4 association strategies that are compared are substantially different in terms of available sample size and statistical power at the present time. While the authors acknowledge the limitations specific to the pQTL dataset, it is generally

true that available exome studies are less well-powered compared to GWAS. What would the results be if the GWAS were downsampled to match the available exome data for each disease? Similarly, eQTL are differentially powered relative to the tissue of interest that is available; the authors utilize the tissue with the strongest association, but that might advantage some disorders (e.g., those primarily expressed in blood) compared to other disorders expressed in tissues less well represented in available eQTL reference datasets.

We acknowledge the difference in sample size and statistical power, and throughout our comparisons we tried to adjust as well as possible individual methods to match each other in terms of sample size, available tissues, and omics entities.

Regarding the difference in sample size between the GWAS and exome data, we conducted sensitivity analyses where we downsampled GWAS to match exome case/control sample sizes. Since we used the UKBB to get exome association results, we used UKBB GWAS (as opposed to consortia GWAS) to conduct fair comparisons between the Exome and GWAS method. While the GWAS method was superior to the Exome method on consortia GWAS data, this was no longer the case when downsampled to UKBB GWAS ($P_{diff} = 0.06$). We made these adjustments for the GWAS method more explicit throughout the manuscript to avoid confusion.

In the second paragraph of the result section “Overview of the analysis”, we made the following changes:

If not specified otherwise, the eQTL-GWAS method refers to the tissue-wide analysis in which the eQTLGen and GTEx data are combined by considering the tissue for which the MR effect was the most significant (Methods). While the GWAS and QTL-GWAS methods focus on common genetic variants, the Exome method considers only rare variants from WES data with minor allele frequencies (MAF) below 1%. Gene scores were based on gene burden tests that aggregate putative loss-of-function and missense variants, and we used the resulting p-values from the WES analysis in the UKBB \cite{backman2021exome}. To allow for a fair comparison with the Exome method while also exploiting disease-specific consortia GWAS summary statistics with maximized case counts, we calculated gene prioritization scores for the GWAS and QTL-GWAS methods using both consortia GWAS and UKBB GWAS data that matched Exome sample sizes (Table S1-3; Methods).

In the Result section “Enrichment of prioritized genes for drug targets” where we present our main results, we more clearly define the rationale behind conducting sensitivity analyses on UKBB GWAS:

Since enrichment results can differ widely across traits and reference databases, we calculated overall enrichment and AUC values across traits and drug databases, including sensitivity analyses restricting GWAS to UKBB data to match Exome sample sizes and common background genes (Table S9, Figure S4; Methods). The overall ORs were 2.17 (UKBB: 1.72), 2.04 (UKBB: 1.67), 1.81 and 1.31 (UKBB: 1.30) for the GWAS, eQTL-GWAS, Exome and pQTL-GWAS methods, respectively.

Regarding the differences in power for different eQTL tissues, we now conducted an additional sensitivity analysis where we exclude the eQTLGen dataset which by far has the largest sample size ($N = 31,684$ compared to $N = 65-573$ for GTEx). This would disadvantage disorders for which relevant genes are primarily expressed in blood and overall lower enrichment values are therefore expected.

We added these results to the Supplementary Table 9 and report them in the result section “Enrichment of prioritized genes for drug targets” which now reads as follows:

Judging by the AUC values, GWAS performed significantly better than eQTL-GWAS ($P_{diff} = 3.1e-5$) also when only considering testable eQTL genes ($P_{diff} = 2.9e-4$). When excluding eQTLGen from the tissue-wide eQTL-GWAS, the performance of eQTL-GWAS slightly dropped (AUC of 52.2% compared to 52.8%; $P_{diff} = 0.019$). Significantly higher AUC values were obtained for the GWAS compared to Exome on consortia data ($P_{diff} = 2.2e-4$) which was no longer the case on UKBB data ($P_{diff} = 0.06$). The difference between eQTL-GWAS and Exome was not significant on either dataset (P_{diff} of 0.12 and 0.77 on consortia and UKBB data, respectively). The number of testable genes was much lower for the pQTL-GWAS method (~1,870 genes). With this set of background genes, GWAS still scored a higher overall AUC (55.1%, $P_{diff} = 2.1e-3$). No difference was observed between the pQTL-GWAS and tissue-wide or whole blood eQTL-GWAS methods (P_{diff} of 0.66 and 0.87, respectively).

Referees' report, second round of review

Reviewer 1

The authors have adequately addressed my comments.

Authors' response to the second round of review

N/A